# Bandit Learning with General Function Classes: Heteroscedastic Noise and Variance-dependent Regret Bounds

## Abstract

We consider learning a stochastic bandit model, where the reward function belongs to a general class of uniformly bounded functions, and the additive noise can be heteroscedastic. Our model captures contextual linear bandits and generalized linear bandits as special cases. While previous works (Kirschner & Krause, 2018; Zhou et al., 2021) based on weighted ridge regression can deal with linear bandits with heteroscedastic noise, they are not directly applicable to our general model due to the curse of nonlinearity. In order to tackle this problem, we propose a *multi-level learning* framework for the general bandit model. The core idea of our framework is to partition the observed data into different levels according to the variance of their respective reward and perform online learning at each level collaboratively. Under our framework, we first design an algorithm that constructs the variance-aware confidence set based on empirical risk minimization and prove a variance-dependent regret bound. For generalized linear bandits, we further propose an algorithm based on follow-the-regularized-leader (FTRL) subroutine and online-to-confidence-set conversion, which can achieve a tighter variance-dependent regret under certain conditions.

## 1 Introduction

Over the past decade, stochastic bandit algorithms have found a wide variety of applications in online advertising, website optimization, recommendation system and many other tasks (Li et al., 2010; McInerney et al., 2018). In the model of stochastic bandits, at each round, an agent selects an action and observes a noisy evaluation of the reward function for the chosen action, aiming to maximize the sum of the received rewards. A general reward function governs the reward of each action from the eligible action set.

A common assumption used in stochastic bandit problems is that the observation noise is conditionally independent and satisfies a uniform tail bound. In real-world applications, however, the variance of observation noise is likely to be dependent on the evaluation point (chosen action) (Kirschner & Krause, 2018). Moreover, due to the dynamic environment in reality, the variance of each action may also be different at each round. This motivates the studies of bandit problems with heteroscedastic noise. For example, Kirschner & Krause (2018) introduced the heteroscedastic noise setting where the noise distribution is allowed to depend on the evaluation point. They proposed *weighted least squares* to estimate the unknown reward function more accurately in the setting where the underlying reward function is linear or lies in a separable Hilbert space (Section 5, Kirschner & Krause 2018).

In this paper, we consider a general setting, where the unknown reward function belongs to a known general function class $\mathcal{F}$ with bounded eluder dimension (Russo & Van Roy, 2013). This captures multi-armed bandits, linear contextual bandits (Abbasi-Yadkori et al., 2011) and generalized linear bandits (Filippi et al., 2010) simultaneously. Since *weighted least squares* highly depends on the linearity of the function class, we propose a *multi-level learning* framework for our general setting. The underlying idea of the framework is to partition the observed data into various levels according to the variance of the noise. The agent then estimates the reward function at each level independently and then exploit all the levels when selecting an action at each round.

While previous work by Kirschner & Krause (2018) considered sub-Gaussian noise with nonuniform variance proxies, we only assume nonuniform variances of noise (Zhou et al., 2021; Zhang et al., 2021), which brings a new challenge of exploiting the variance information of the noise to obtain tighter *variance-aware confidence sets*.

Under our *multi-level learning* framework, we first design an algorithm based on empirical risk minimization and Optimism-in-the-Face-of-Uncertainty (OFU) principle, and prove a variance-dependent regret bound. For a special class of bandits namely generalized linear bandits with heteroscedastic noise, we further propose an algorithm using follow-the-regularized-leader (FTRL) as an online regression subroutine and adopting the technique of online-to-confidence-set conversion (Abbasi-Yadkori et al., 2012; Jun et al., 2017). This algorithm achieves a provaly tighter regret bound when the range of the reward function is relatively wide compared to the magnitude of noise.

Our main contributions are summarized as follows:

- We develop a new framework called *multi-level regression*, which can be applied to heteroscedastic bandits, even when the reward function class does not lie in a separable Hilbert space.

- Under our framework, we design tighter *variance-aware upper confidence bounds* for bandits with general reward functions, and propose an bandit learning algorithm based on empirical risk minimzation. We show that our algorithm enjoys variance-dependent regret upper bounds which can be regarded as a strict extension of previous algorithms which obtain variance-dependent regret bounds on simpler bandit models (Zhou et al., 2021; Zhang et al., 2021).

- For generalized linear bandits (Filippi et al., 2010; Jun et al., 2017), which is a special case of our model class, we further propose an algorithm based on online-to-confidence-set conversion. We first prove a variance-dependent regret bound for follow-the-regularized-leader (FTRL) for the online regression problem derived from generalized linear function class, and then convert the online learning regret bound to the bandit learning confidence set. We show that our algorithm can achieve a tighter regret bound for generalized linear bandits.

- As a by-product, our regret bound for FTRL improves the state-of-the-art regret result $\widetilde{O}(d^2 R^2)$ obtained by stochastic online linear regression (Ouhamma et al., 2021) to $\widetilde{O}(d\sigma_{\max}^2)$ (omitting the terms without dependence on $d$), where $d$ is the dimension of contexts, $R$ is the upper bound of the sub-Gaussian norm of the noises at each step, and $\sigma_{\max}$ is the upper bound of the variances of the noises.

Table 1: A summary of our regret results and previous results under different settings.

| Function Class | Algorithm | Regret | Efficiency | Unknown VAR |
|---|---|---|---|---|
| Linear | Weighted OFUL (Zhou et al., 2021) | $\widetilde{O}\left(d\sqrt{J} + R\sqrt{dT}\right)$ | Computationally efficient | |
| Linear | VOFUL (Zhang et al., 2021) | $\widetilde{O}\left(d^{4.5}\sqrt{J} + d^5 R\right)$ | Inefficient | ✓ |
| General | ML$^2$ + ERM (This work) | $\widetilde{O}(\sqrt{\dim_E \log(N_\alpha)J} + \sqrt{\dim_E \log(N_\alpha)RT}$ | Oracle efficient | |
| G-Lin | ML$^2$ + GLOC (This work) | $\widetilde{O}(\frac{K}{\kappa}d\sqrt{J} + \frac{K}{\kappa}(KAB + R)\sqrt{dT})$ | Computationally efficient | |

Refer to Section 3 for the definitions of $\dim_E, \sigma_t, J$ and $R$, Section 6 for the definitions of $\kappa, K, A, B$. We write general function class with eluder dimension $\dim_E$ as 'General' and generalized linear function class as 'G-Lin' for short. Oracle efficiency refers to the computational efficiency given a regression oracle (i.e., empirical risk minimization) for the involved function class and an optimization oracle which maximizes the reward function $f(x)$ for a fixed $x$ under some constraint set of $f$.

**Notation.** We use lower case letters to denote scalars, and use lower and upper case bold face letters to denote vectors and matrices respectively. We denote by $[n]$ the set $\{1, \ldots, n\}$. For a vector $\mathbf{x} \in \mathbb{R}^d$ and matrix $\mathbf{\Sigma} \in \mathbb{R}^{d \times d}$, a positive semi-definite matrix, we denote by $\|\mathbf{x}\|_2$ the vector's Euclidean norm and define $\|\mathbf{x}\|_{\mathbf{\Sigma}} = \sqrt{\mathbf{x}^\top \mathbf{\Sigma} \mathbf{x}}$. For two positive sequences $\{a_n\}$ and $\{b_n\}$ with $n = 1, 2, \ldots$, we write $a_n = O(b_n)$ if there exists an absolute constant $C > 0$ such that $a_n \leq Cb_n$ holds for all $n \geq 1$ and write $a_n = \Omega(b_n)$ if there exists an absolute constant $C > 0$ such that $a_n \geq Cb_n$ holds for all $n \geq 1$. Let $\mathcal{N}(\mathcal{F}, \alpha, \|\cdot\|_\infty)$ denote the $\alpha$-covering number of $\mathcal{F}$ in the

sup-norm $\| \cdot \|_\infty$. If there is no ambiguity, we may write $\mathcal{N}(\mathcal{F}, \alpha, \| \cdot \|_\infty)$ as $N_\alpha$ for short. We use $\widetilde{O}(\cdot)$ to further hide the polylogarithmic factors other than log-covering numbers.

## 2 RELATED WORK

**Learning with heteroscedastic noise.** Heteroscedastic noise has been studied in many different settings such as active learning Antos et al. (2010), regression (Aitken, 1936; Goldberg et al., 1997; Chaudhuri et al., 2017; Kersting et al., 2007), principle component analysis (Hong et al., 2016; 2018) and Bayesian optimization (Assael et al., 2014). However, only a few works have considered heteroscedastic noise in bandit settings. Cowan et al. (2015) considered a variant of multi-armed bandits where the noise at each round is a Gaussian random variable with unknown variance. Kirschner & Krause (2018) is the first to formally introduce the concept of stochastic bandits with heteroscedastic noise. In their model, the variance of the noise at each round $t$ is a function of the evaluation point $x_t$, $\rho_t = \rho(x_t)$, and they further assume that the noise is $\rho_t$-sub-Gaussian. $\rho_t$ can either be observed at time $t$ or either be estimated from the obsevations. Zhou et al. (2021) considered linear bandits with heteroscedastic noise and generalized the heteroscedastic noise setting in Kirschner & Krause (2018) in the sense that they no longer assume the noise to be $\rho_t$-sub-Gaussian, but only requires the variance of noise to be upper bounded by $\rho_t^2$ and the variances are arbitrarily decided by the environment, which is not necessarily a function of the evaluation point. In the same setting as in Zhou et al. (2021), Zhang et al. (2021) further considered a strictly harder setting where the noise has unknown variance. They proposed an algorithm which can deal with unknown variance through a computationally inefficient clip technique. Our work basically considers the noise setting proposed by Zhou et al. (2021) and further generalizes their setting to bandits with general function classes. We will consider to extend it to the harder setting as Zhang et al. (2021) as future work.

**Bandits with known function classes.** Moving beyond multi-armed bandits, there have been significant theoretical advances on stochastic bandits with function approximation. Among them, there is a huge body of literature on linear bandit problems where the reward function is assumed to be a linear function of the feature vectors attached to the actions (Dani et al., 2008; Abbasi-Yadkori et al., 2011; Chu et al., 2011; Li et al., 2019; 2021b). Generalizing the restrictive linear rewards, there has also been a flurry of studies on generalized linear bandit problems (Filippi et al., 2010; Jun et al., 2017; Li et al., 2017; Kveton et al., 2020).

As for stochastic bandits with general function classes, the seminal work by Russo & Van Roy (2013) introduced the notion of eluder dimension to measure the complexity of the function class and provided a general UCB-like algorithm that works for any given class of reward functions with bounded eluder dimension. They further proved a regret upper bound of order $\widetilde{O}(\sqrt{\dim_E \log \mathcal{N} \cdot T})$ for their proposed algorithm where $\dim_E$ is the eluder dimension and $\log \mathcal{N}$ stands for the log-covering number of the funciton class. Linear bandits and generalized linear bandit problems can be seen as special cases as their proposed general model.

**Online-to-confidence-set conversion.** Abbasi-Yadkori et al. (2012) may be the first one to introduce the technique that takes in an online learning subroutine and turns the output of it into a confidence set at each round. While Abbasi-Yadkori et al. (2012) considered applying this technique in linear bandits, Jun et al. (2017) generalized and introduced the previous approach to *Generalized Linear Online-to-confidence-set Conversion* (GLOC) and applied it to generalized linear bandits.

**Online regression for linear functions.** Online linear regression has long been studied in the setting where the response variables (or labels) are bounded and chosen by an adversary (Bartlett et al., 2015; Cesa-Bianchi et al., 1996; Kivinen & Warmuth, 1997; Littlestone et al., 1991; Malek & Bartlett, 2018), to mention a few. A recent work (Ouhamma et al., 2021) considers the stochastic setting where the response variables are unbounded and revealed by the environment with additional random noise on the true labels. Ouhamma et al. (2021) discussed the limitations of online learning algorithms in the adversarial setting and further advocates for the need of complementary analyses for existing algorithms under stochastic unbounded setting.

## 3 MULTI-LEVEL LEARNING FRAMEWORK

We introduce the Multi-level learning framework in this section.

### 3.1 PRELIMINARIES

**General function class** Following (Russo & Van Roy, 2013), we introduce the $\epsilon$-dependence and eluder dimension notion, which are used to measure the complexity of a general function class $\mathcal{F}$.

**Definition 3.1** ($\epsilon$-dependence, Russo & Van Roy 2013)**.** An action $a \in \mathcal{A}$ is $\epsilon$-dependent on actions $\{a_1, a_2, \cdots, a_n\} \in \mathcal{A}$ with respect to $\mathcal{F}$ if any pair of functions $f, \widetilde{f} \in \mathcal{F}$ satisfying $\sum_{i=1}^{n}(f(a_i) - \widetilde{f}(a_i))^2 \le \epsilon^2$ also satisfies $f(a) - \widetilde{f}(a) \le \epsilon$. Further, $a$ is $\epsilon$-independent of $\{a_1, \cdots, a_n\}$ with respect to $\mathcal{F}$ if $a$ is not $\epsilon$-dependent on $\{a_1, \cdots, a_n\}$.

**Definition 3.2** (eluder dimension, Russo & Van Roy 2013)**.** The $\epsilon$-eluder dimension $\dim_E(\mathcal{F}, \epsilon)$ is the length $d$ of the longest sequence of elements in $\mathcal{A}$ such that, for some $\epsilon' \ge \epsilon$, every element is $\epsilon'$-independent of its predecessors.

In this work we focus on general function class $\mathcal{F}$ with bounded eluder dimension, and we consider generalized linear function class as a special case. We would like to point out that the function class with small eluder dimension is strictly larger than linear and generalized linear bandits (Li et al., 2021a), while neural networks with ReLU activation do not have a small eluder dimension (their eluder dimension has an exponential dependence on the input dimension) (Dong et al., 2021). We leave it as future work to consider even more general function classes.

**Definition 3.3** (width)**.** Let $w_{\widetilde{\mathcal{F}}}(a) = \sup_{\underline{f}, \overline{f} \in \widetilde{\mathcal{F}}} \left( \overline{f}(a) - \underline{f}(a) \right)$. For an action set $\widetilde{\mathcal{A}} \subseteq \mathcal{A}$, we use $w_{\widetilde{\mathcal{F}}}(\widetilde{\mathcal{A}})$ to denote $\sup_{a \in \widetilde{\mathcal{A}}} w_{\widetilde{\mathcal{F}}}(a)$.

To deal with infinite or continuous action sets, we make the following assumption that the reward function class is known in advance by the agent. Notice that for the finite multi-armed bandit case, we can choose $\mathcal{F}$ as the set that includes all the eligible functions.

**Assumption 3.4** (Known Reward Function Class)**.** The unknown reward function $f^*$ belongs to an accessible function class $\mathcal{F} = \{f_\theta : \mathcal{A} \to \mathbb{R} | \theta \in \Theta\}$.

**Bandit models** We consider a heteroscedastic variant of the classic stochastic bandit problem with general function classes. At each round $t \in [T]$ ($T \in \mathbb{N}$), the agent observes a decision set $\mathcal{D}_t \subseteq \mathcal{A}$ which is chosen by the environment. The agent then selects an action $a_t \in \mathcal{D}_t$ and observes reward $r_t$ together with a corresponding variance upper bound $\sigma_t^2$. We assume that $r_t = f^*(a_t) + \epsilon_t$ where $f^* : \mathcal{A} \to \mathbb{R}$ is an underlying real-valued reward function which is unknown to the learner and $\epsilon_t$ is a random noise. We make the following assumption on $\epsilon_t$.

**Assumption 3.5.** For each $t$, $\epsilon_t$ satisfies that $\epsilon_t | a_{1:t}, \epsilon_{1:t-1}$ is a $R$-sub-Gaussian random variable ($R > \sigma_t$) and $\mathbb{E}[\epsilon_t | a_{1:t}, \epsilon_{1:t-1}] = 0$, $\mathbb{E}[\epsilon_t^2 | a_{1:t}, \epsilon_{1:t-1}] \le \sigma_t^2 := \sigma_t^2(a_{1:t}, r_{1:t-1})$.

where $\sigma_t$ can be either a constant or a random variable dependent on $a_{1:t}$ and $r_{1:t-1}$.

**Remark 3.6.** $\sigma_t$ can be seen as either a given information from the environment, or an estimator of the noise variance at $t$-th round based on all past observations, as discussed in the information directed sampling bandit (Kirschner & Krause, 2018). For instance, we consider a bandit problem with a two-point action distribution, where the variance of an action can be estimated through the estimation of the mean of the action. Similar estimation procedure has been studied in Lattimore et al. (2015). The details are in Appendix C. We can further consider the MDP setting. With a confidence set $\mathcal{P}$ that includes the true transition dynamic, the conditional variances of the value functions $\bar{V}$ at state $s$ and action $a$ can be estimated by $\sup_{p \in \mathcal{P}} \left\{ \sum_{s' \in S} p(s|s,a) \bar{V}^2(s') - \left[ \sum_{s' \in S} p(s'|s,a) \bar{V}(s') \right]^2 \right\}$. So the variances of value functions can be efficiently estimated for the MDP setting.

For simplicity, let $J = \sum_{t=1}^{T} \sigma_t^2$. This assumption on $\epsilon_t$ is a slightly generalized version of that in Zhou et al. (2021) in the sense that the noise is not necessarily bounded by $R$. The goal of the agent is to minimize the following cumulative regret:

$$\text{Regret}(T) := \sum_{t=1}^{T} [f^*(a_t^*) - f^*(a_t)], \qquad (3.1)$$

where the optimal action $a_t^*$ at round $t \in [T]$ is defined as $a_t^* := \operatorname{argmax}_{a \in \mathcal{D}_t} f^*(a)$.

---

**Algorithm 1** ML$^2$ with OFU principle

---

1: **Input:** $T, \mathcal{A}, \mathcal{F}, R, \overline{\sigma} > 0$.
2: **Initialize:** Set $L \leftarrow \lceil \log_2 R/\overline{\sigma} \rceil$ and $\mathcal{C}_{1,l} \leftarrow \mathcal{F}, \Psi_{1,l} \leftarrow \varnothing$ for all $l \in [L]$.
3: **for** $t = 1 \cdots T$ **do**
4:     Observes $\mathcal{D}_t$.
5:     Choose action $a_t = \text{argmax}_{a \in \mathcal{D}_t} \min_{l \in [L]} \max_{f \in \mathcal{C}_{t,l}} f(a)$.
6:     Observe stochastic reward $r_t$ and $\sigma_t^2$.
7:     Find $l_t$ such that $2^{l_t+1}\overline{\sigma} \geq \max(\overline{\sigma}, \sigma_t) \geq 2^{l_t}\overline{\sigma}$.
8:     Update $\Psi_{t+1,l_t} \leftarrow \Psi_{t,l_t} \cup \{t\}$ and $\Psi_{t+1,l} \leftarrow \Psi_{t,l}$ for all $l \in [L]\backslash\{l_t\}$.
9:     Update $\mathcal{C}_{t+1,l}$ according to $\Psi_{t+1,l}$ through a regression subroutine (e.g., Algorithm 2).
10: **end for**

---

## 3.2    Multi-level Learning framework

**Existing approach.** To tackle the heteroscedastic bandit problem, for the case where the $\mathcal{F}$ is the linear function class (i.e., $f(a) = \langle \boldsymbol{\theta}^*, a \rangle$ for some $\boldsymbol{\theta}^* \in \mathbb{R}^d$), a *weighted linear regression* framework (Kirschner & Krause, 2018; Zhou et al., 2021) has been proposed. Generally speaking, at each round $t \in [T]$, weighted linear regression constructs a confidence set $\mathcal{C}_t$ based on the empirical risk minimizarion (ERM) for all previous observed actions $a_s$ and rewards $r_s$ as follows:

$$\boldsymbol{\theta}_t \leftarrow \underset{\boldsymbol{\theta} \in \mathbb{R}^d}{\text{argmin}} \, \lambda\|\boldsymbol{\theta}\|_2^2 + \sum_{s \in [t]} w_s(\langle \boldsymbol{\theta}, a_s \rangle - r_s)^2, \quad \mathcal{C}_t \leftarrow \left\{ \boldsymbol{\theta} \in \mathbb{R}^d \,\middle|\, \sum_{s=1}^t w_s(\langle \boldsymbol{\theta}, a_s \rangle - \langle \boldsymbol{\theta}_t, a_s \rangle)^2 \leq \beta_t \right\},$$

where $w_s$ is the weight, and $\beta_t, \lambda$ are some parameters to be specified. $w_s$ is selected in the order of the inverse of the variance $\sigma_s^2$ at round $s$ to let the variance of the rescaled reward $\sqrt{w_s}r_s$ upper bounded by 1. Therefore, after the weighting step, one can regard the heteroscedastic bandits problem as a homoscedastic bandits problem and apply existing theoretical results to it. To deal with the general function case, a direct attempt is to replace the $\langle \boldsymbol{\theta}, a \rangle$ appearing in above construction rules with $f(a)$. However, such an approach requires that $\mathcal{F}$ is *close* under the linear mapping, which does not hold for general function class $\mathcal{F}$.

**Multi-level Learning framework (ML$^2$).** To deal with the nonlinearity issue, we propose a novel framework $ML^2$ in Algorithm 1. At the core of our design is the idea of partitioning the observed data into several levels and 'packing' data with similar variance upper bounds into the same level as shown in line 7-8 of Algorithm 1. Note that we use a small real number $\overline{\sigma}$ to ensure that the number of levels is bounded. Specifically, for any two data belong to the same level with variaces larger than $\overline{\sigma}$, their variance will be at most twice larger than the other. Next in line 9, our framework calls a subroutine to estimate $f^*$ according to the data points in $\Psi_{t+1,l}$. Since the variances of the data in the same level are nearly the same, we let algorithms that work for homoscedastic bandit problem run on the data in the same level. Particularlly, we use the Empirical risk minimization (ERM) algorithm as described in Algorithm 2 for Sections 4 and 5. In Section 6, we show the power of using Algorithm 4 as the regression subroutine. Then in line 5, the agent makes use of $L$ confidence sets simultaneously to select an action based on the *optimism-in-the-face-of-uncertainty* (OFU) principle over all $L$ levels. More specifically, the algorithm chooses the action opmimisticly according to each confidence set but inclines to select the confidence set with the most pessimistic evaluation. In the following sections, we will consider several different settings to show the power of ML$^2$.

## 4    Warmup: noise with additional sub-Gaussian assumption

We first consider a simplified variant of our problem, where each noise is also sub-Gaussian.

**Assumption 4.1** (Sub-Gaussianity of noise). $\epsilon_t$ is conditionally $\sigma_t$-sub-Gaussian on $a_{1:t}, \epsilon_{1:t-1}$.

Such a sub-Gaussian assumption on noise has been considered by Kirschner & Krause (2018). Next we show the regret upper bound for ML$^2$ with ERM. For simplicity, in the following results, let $\dim_E$ denote $\dim_E(\mathcal{F}, 1/T^2)$.

**Theorem 4.2** (Gap-independent regret bound for bandits with heteroscedastic sub-Gaussian noise). Suppose Assumption 3.4 and 4.1 hold and $|f^*(a)| \leq C$ for all $a \in \mathcal{A}$. For all $t \in [T], l \in [L]$ and $\delta \in (0,1), \alpha > 0, \overline{\sigma} > 0$, if we apply Algorithm 2 as a subroutine of Algorithm 1 (in line 9) and set

---

**Algorithm 2** Empirical risk minimization (ERM) for partitioned data

---

1: **Input:** Level $l$, time $t$ and set of data points $\Psi_{t+1,l}$.

2: Compute $\widehat{f}_{t+1,l} \leftarrow \operatorname{argmin}_{f \in \mathcal{F}} \sum_{s \in \Psi_{t+1,l}} \left( f(a_s) - r_s \right)^2$.

3: **Return** $\mathcal{C}_{t+1,l} \leftarrow \left\{ f \in \mathcal{F} \mid \sum_{s \in \Psi_{t+1,l}} \left[ f(a_s) - \widehat{f}_{t+1,l}(a_s) \right]^2 \leq \beta_{t+1,l} \right\}$

---

$\beta_{t,l}$ as the square root of

$$8(2^{l+1} \cdot \overline{\sigma})^2 \log(2N_\alpha L/\delta) + 4t\alpha\left(C + \sqrt{(2^{l+1} \cdot \overline{\sigma})^2 \log(4t(t+1)L/\delta)}\right), \tag{4.1}$$

where $N_\alpha = \mathcal{N}(\mathcal{F}, \alpha, \|\cdot\|_\infty)$ and $L = \lceil \log_2 R/\overline{\sigma} \rceil$ (recall the definition of $L$ in Algorithm 1), then with probability at least $1 - \delta$, the regret for the first $T$ rounds is bounded as follows:

$$\text{Regret}(T) \leq L + 2C \dim_E L + 8\sqrt{2L \dim_E (J + \overline{\sigma}^2 T) \log(2N_\alpha L/\delta)}$$

$$+ 4\sqrt{L \dim_E \alpha}\sqrt{C + 2R\sqrt{\log(4T(T+1)L/\delta)}}T.$$

**Corollary 4.3.** Let the same conditions as in Theorem 4.2 hold. Set $\alpha = T^{-2}$ and $\overline{\sigma} = \dim_E^{-1}(\log(2N_\alpha L/\delta)\sqrt{T})^{-1}$. Then with probability at least $1 - \delta$, when $T$ is large enough, the regret for the first $T$ rounds is bounded as $\text{Regret}(T) = \widetilde{O}\left(\sqrt{\dim_E \log(\mathcal{N}(\mathcal{F}, T^{-2}, \|\cdot\|_\infty))J}\right)$.

**Remark 4.4.** Our result is strictly tighter than the $\widetilde{O}\left(R\sqrt{\dim_E \log(\mathcal{N}(\mathcal{F}, T^{-2}, \|\cdot\|_\infty))T}\right)$ regret achieved by Russo & Van Roy (2013) since $J = \sum_{t=1}^{T} \sigma_t^2 \leq R^2 T$. In the worst case, when $\sigma_1 = \cdots = \sigma_T = R$, our result degrades to their result. Our improvement in regret is due to the utilization of variance information. When the variance information is provided or can be estimated, our algorithm can achieve better regrets for bandits with general function classes studied in this paper, while existing algorithms cannot.

**Remark 4.5.** When restricted to linear contextual bandits with dimension $d$, since $\log \mathcal{N}(\mathcal{F}, T^{-2}, \|\cdot\|_\infty) = \widetilde{O}(d)$, $\dim_E = \widetilde{O}(d)$ (Russo & Van Roy, 2013), our result can be written as $\widetilde{O}(d\sqrt{J})$, which matches the result of using weighted linear ridge regression for heteroscedastic linear bandit under our assumptions on noise (Kirschner & Krause, 2018; Zhou et al., 2021).

We also provide a gap-dependent regret bound for general function class setting in section D, generalizing the previous gap-dependent regret bound in linear bandits (Abbasi-Yadkori et al., 2011).

## 5 General results for bandits with heteroscedastic noise

In this section, we consider the original setting introduced in Section 3 without Assumption 4.1. In the following subsections, we will show that Algorithm 2 still works with refined value of $\beta$.

### 5.1 Variance-aware confidence set

In this genral setting, directly applying the confidence set used in the previous work (Russo & Van Roy, 2013) gives no improvement since our confidence sets $\mathcal{C}_{t,l}$ do not adopt the variance information. We show in the following theorem that our new designed $\mathcal{C}_{t,l}$ with a new confidence radius $\beta$ still ensures that the confidence set is large enough to contain $f^*$ with high probability, and exploits the variance information at the same time.

**Theorem 5.1** (Variance-dependent confidence sets). Suppose that $|f^*(a)| \leq C$ for all $a \in \mathcal{A}$. For any $\alpha > 0$ and $\delta \in (0, 1/2)$, if we set $\beta_{t,l}$ as the square root of

$$12C\alpha t + 4\alpha \overline{R} t + 8/3 \cdot C\overline{R} \log(2N_\alpha t^2/\delta) + 16 \cdot (2^{l+1}\overline{\sigma})^2 \log(2N_\alpha t^2/\delta),$$

where $\overline{R} = R\sqrt{2 \log(4t^2/\delta)}$, $N_\alpha = \mathcal{N}(\mathcal{F}, \alpha, \|\cdot\|_\infty)$, then $f^* \in \mathcal{C}_{t,l}$ with probability at least $1 - 2\delta$ for any fixed $t, l$.

**Remark 5.2.** With a small $\alpha$, we have $\beta_{t,l} = \widetilde{O}(2^{2l}\overline{\sigma}^2 \log N_\alpha + CR \log N_\alpha)$. Compared with the corresponding previous result $\widetilde{O}(R^2 \log N_\alpha)$ (Russo & Van Roy, 2013; Ayoub et al., 2020), our confidence set is tighter when $C$ is relatively small compared to $R$.

## 5.2 REGRET UPPER BOUNDS FOR ML$^2$ WITH ERM

We derive our general results with the variance-aware confidences sets described in the last subsection. In this part, we write $\dim_E(\mathcal{F}, T^{-1})$ as $\dim_E$ for short.

**Theorem 5.3** (Gap-independent regret bound for bandits with heteroscedastic noise). Suppose Assumption 3.4 holds and $|f^*(a)| \leq 1$ for all $a \in \mathcal{A}$. For all $t \in [T], l \in [L]$ and $\delta \in (0,1), \alpha > 0, \overline{\sigma} > 0$, if we apply Algorithm 2 as a subroutine of Algorithm 1 (in line 9) and set $\beta_{t,l}$ as the square root of $12\alpha t + 4\alpha \overline{R} t + 8/3 \cdot \overline{R} \log(2N_\alpha t^2 L/\delta) + 16 \cdot (2^{l+1}\overline{\sigma})^2 \log(2N_\alpha t^2 L/\delta)$, where $L = \lceil \log_2 R/\overline{\sigma} \rceil$, $N_\alpha = \mathcal{N}(\mathcal{F}, \alpha, \|\cdot\|_\infty)$ and $\overline{R} = R\sqrt{2\log(4t^2 L/\delta)}$ (with a slight abuse of notation), then with probability at least $1 - 2\delta$, the regret for the first $T$ rounds is bounded as follows:

$$\text{Regret}(T) \leq \sqrt{L}\left(2\sqrt{\dim_E T} + 1\right) + 4\sqrt{L \dim_E(\log T + 1)\alpha}\sqrt{3 + \overline{R}T}$$

$$+ 2\sqrt{\frac{8}{3}L \dim_E(\log T + 1)\overline{R}\log(2N_\alpha t^2 L/\delta)T} + 16\sqrt{L \dim_E(\log T + 1)\log(2N_\alpha T^2 L/\delta)}\sqrt{J + T\overline{\sigma}^2}.$$

**Corollary 5.4.** Assume $R = \Omega(1)$. Let the same conditions as in Theorem 5.3 hold. Set $\alpha = T^{-2}, \overline{\sigma} = 1$. Then with probability at least $1 - \delta$, when $T$ is large enough, the regret for the first $T$ rounds is bounded as $\text{Regret}(T) = \widetilde{O}\left(\sqrt{\dim_E \log N_\alpha J} + \sqrt{R \dim_E \log N_\alpha T}\right)$.

**Remark 5.5.** Compared with the result shown in Corollary 4.3, the additional term of order $\widetilde{O}(\sqrt{\dim_E \log N_\alpha RT})$ is due to a larger confidence set by the absence of Assumption 4.1.

**Remark 5.6.** When restricted to heteroscedastic linear contextual bandits of dimension $d$, our regret bound can be written as $\widetilde{O}(d\sqrt{J} + \sqrt{R}d\sqrt{T})$. With a slightly more restricted assumption on noise, Zhou et al. (2021) achieved a result of order $\widetilde{O}(d\sqrt{J} + R\sqrt{dT})$. Our result is appealing when the sub-Gaussian parameter of noise $R$ is much larger than 1 (or the range of the reward function, equivalently). When $R$ is small, our result becomes sub-optimal due to the property of our variance-aware confidence set.

We also provide a gap-dependent regret bound in Appendix D.

## 6 TIGHTER BOUNDS FOR GENERALIZED LINEAR BANDITS

Our general result shown in Theorem 5.3 has an additional term of order $\widetilde{O}(\sqrt{\dim_E \log N_\alpha RT})$ which makes the result sub-optimal when $R$ is close to the range of the reward function. In this section, we consider a special case, generalized linear bandits with heteroscedastic noise. We show how to get rid of the $\widetilde{O}(\sqrt{\dim_E \log N_\alpha RT})$ term in the upper bound of the regret, and achieve a better result when $R$ is relatively small or close to the bound of the reward function.

### 6.1 GENERALIZED LINEAR BANDITS

Following Filippi et al. (2010); Jun et al. (2017), we consider the generalized linear function class defined as follows.

**Assumption 6.1** (Generalized linear function class). Action set $\mathcal{A}$ and $\Theta$ in Assumption 3.4 are subsets of $\mathbb{R}^d$. There exists a known link function $h$, such that $\forall \mathbf{a} \in \mathcal{A}$ and $f_{\boldsymbol{\theta}} \in \mathcal{F}$, $f_{\boldsymbol{\theta}}(\mathbf{a}) = h(\boldsymbol{\theta}^\top \mathbf{a})$. Let $f^* = f_{\boldsymbol{\theta}^*}$. Assume that $\|\boldsymbol{\theta}^*\|_2 \leq B$, $\sup_{\mathbf{a} \in \mathcal{A}} \|\mathbf{a}\|_2 \leq A$.

To make the problem tractable, we need the following assumption on $h$.

**Assumption 6.2** (Assumption 1, Jun et al. 2017). $h$ is $K$-Lipschitz on $[-A \cdot B, A \cdot B]$ and continuously differentiable on $(-A \cdot B, A \cdot B)$. Furthermore, $\inf_{z \in (-A \cdot B, A \cdot B)} h'(z) = \kappa$ for some $\kappa > 0$.

Next we propose the follow-the-regularized-leader (FTRL) framework (Shalev-Shwartz & Singer, 2007; Xiao, 2010; Hazan, 2019) in Algorithm 3, which is the key component of our final algorithm. Note that when dealing with bandit setting, we maintain an independent process executing Algorithm 3 for each variance level instead of feeding all the data points into a single FTRL online learner. Here we number the data points with $t = 1, 2, \cdots$ for simplicity with a slight abuse of notation. In Algorithm 3 we will use a loss function $\ell$ and a regularized function $\phi$. $\phi$ is defined as

---

**Algorithm 3** Follow The Regularized Leader (FTRL)

---

1: **Input:** $\mathcal{F}, R$.
2: **for** $t \geq 1$ **do**
3:     Output $\boldsymbol{\theta}_t \leftarrow \operatorname{argmin}_{\boldsymbol{\theta} \in \mathbb{R}^d} \phi(\boldsymbol{\theta}) + \sum_{s=1}^{t-1} \ell(\boldsymbol{\theta}^\top \mathbf{a}_s, r_s)$.
4:     Observe $\mathbf{a}_t, r_t$.
5: **end for**

---

$\phi(\theta) = c \cdot \|\theta\|_2^2$ where $c$ is a constant which will be specified later in Theorem 6.5. To align with our Assumption 6.1, we select the loss function as follows, following Jun et al. (2017):

**Assumption 6.3** (Loss function, Jun et al. 2017). The loss function $\ell$ in Algorithm 3 is selected as follows: $\ell(z, r) = -rz + m(z)$, $\ell_t(\boldsymbol{\theta}) = \ell(\boldsymbol{\theta}^\top \mathbf{a}_t, r_t)$, where $m(z)$ satisfies $m'(z) = h(z)$.

### 6.2 VARIANCE-DEPENDENT REGRET FOR FTRL

Before proposing our final algorithm for generalized linear bandits, we first propose a variance-dependent complexity result for FTRL, since it is already nontrivial and reveals some interesting properties about our setting. We define a notion of *regret of online regression*, named by $\mathrm{reg}_t$, as follows. The concept of regret of online regression has been introduced in the previous work (Abbasi-Yadkori et al., 2012; Jun et al., 2017). In detail, it is used to characterize the complexity for FTRL to learn the generalized linear function.

**Definition 6.4.** Let $\mathrm{reg}_t = \sum_{s=1}^t \ell(\mathbf{a}_s^\top \boldsymbol{\theta}_s, r_s) - \sum_{s=1}^t \ell(\mathbf{a}_s^\top \boldsymbol{\theta}^*, r_s)$.

Our definition of regret for online regression is slightly different that in prior works (Abbasi-Yadkori et al., 2012; Jun et al., 2017). Here $\boldsymbol{\theta}^*$ is choosen to be the true parameter for the bandit model, while $\boldsymbol{\theta}^*$ is often chosen as $\arg\inf_{\boldsymbol{\theta} \in \Theta} \ell_s(\boldsymbol{\theta})$ in Abbasi-Yadkori et al. (2012); Jun et al. (2017). From the perspective of online learning, the algorithms and the corresponding analyses are usually introduced for either the realizable setting where there exists an underlying $\boldsymbol{\theta}^*$ that incurs zero loss, or the adversarial setting where the bounded label $r_s$ in each round $s$ can be arbitrarily chosen by the adversary. As a result, the previous approaches by Abbasi-Yadkori et al. (2012); Jun et al. (2017) do not exploit the 'stochastic' property of the labels. Provided that the labels are sequentially generated with additional stochastic noise, our definition is more reasonable and natural. A recent work focusing on stochastic online linear regression also discussed the limitation of adversarial setting (Section 2.2, Ouhamma et al. 2021).

Next we propose a bound for $\mathrm{reg}_t$ which adopts the variance information.

**Theorem 6.5** (Regret of FTRL). Set $\phi(\boldsymbol{\theta}) = 2A^2 K^2 \|\boldsymbol{\theta}\|_2^2 / \kappa$ and assume that all the data points fed into the algorithm are of noise variance bounded by $\sigma_{\max}^2$, then with probability at least $1 - 3\delta$, $\forall t \geq 1$, the regret of Algrithm 3 for the first $t$ rounds is bounded as follows:

$$\mathrm{reg}_t \leq \frac{8A^2 K^2 B^2}{\kappa} + \frac{9}{2\kappa} R^2 \log^2(4t^2/\delta) + 3 \frac{\sigma_{\max}^2}{\kappa} d \log\left(1 + \frac{tA\kappa^2}{4dK^2}\right).$$

**Remark 6.6.** Jun et al. (2017) analyzed the online learning regret for the same function class and loss function with our setting. Their result yields a $\mathrm{reg}_t$ in the order of $\widetilde{O}(\frac{K^2 A^2 B^2 + R^2}{\kappa} d)$. Our result improves their result in two aspects. First, $R$ is strictly larger than $\sigma_{\max}$ since a $R$-sub-Gaussian random variable is definitely of variance lower than $R^2$. Second, when we consider cases where the bound of reward functions (i.e., $KAB$) is extremely large compared to $R$, their result becomes $\widetilde{O}(K^2 A^2 B^2 d / \kappa)$, which has an additional linear dependence on $d$.

**Remark 6.7.** Consider a special case where $\kappa = K = 1$. Our result degrades to $\widetilde{O}(A^2 B^2 + R + \sigma_{\max}^2 d)$. This is essentially a regret upper bound for stochastic online linear regression with square loss. Recently Ouhamma et al. (2021) studied this stochastic setting and managed to get rid of the $\widetilde{O}(A^2 B^2 d)$ term in classic result for online linear regression considering adversarial setting. Ouhamma et al. (2021) derived a high probability regret bound of $\widetilde{O}(R^2 d^2)$ after omitting the $o(\log(T)^2)$ terms (Theorem 3.3, Ouhamma et al. 2021). Unlike their result, our result does not suffer from the quadratic dependence on $d$, and our result depends on $\sigma_{\max}^2 d$ rather than $R^2 d^2$. Therefore, our result is better than that in Ouhamma et al. (2021) when $d$ is large. We also notice that the discussion in Sec. 3.3 in Ouhamma et al. (2021) yields an improved expected regret of order $O(R^2 d \log^2 T + R^2 d^2 \log T \log\log T)$, but the first term dominates the regret in the asymptotic sense, i.e., only when $T$ is very large ($T \geq (\log T)^d$).

---

**Algorithm 4** GLOC with multi-level FTRL learners

---

1: **Initialize:** $\overline{\mathbf{V}}_{0,l} \leftarrow \lambda \mathbf{I}$ for all $l \in [L]$.
2: **while input** $t, l_t, \mathbf{a}_t, r_t$ **do**
3:     Set $\boldsymbol{\theta}_{t,l_t}$ following Algorithm 3, where $\boldsymbol{\theta}_{t,l_t} \leftarrow \operatorname{argmin}_{\boldsymbol{\theta} \in \mathbb{R}^d} \phi(\boldsymbol{\theta}) + \sum_{s \in \Psi_{t+1,l_t}} \ell(\boldsymbol{\theta}^\top \mathbf{a}_s, r_s)$.

4:     Find $t' = \max \Psi_{t,l_t}$.
5:     Update $\overline{\mathbf{V}}_{t,l_t} \leftarrow \overline{\mathbf{V}}_{t',l_t} + \mathbf{a}_t \mathbf{a}_t^\top$, $z_{t,l_t} \leftarrow \mathbf{a}_t^\top \boldsymbol{\theta}_{t,l_t}$.
6:     Compute $\widehat{\boldsymbol{\theta}}_{t,l_t} \leftarrow \overline{\mathbf{V}}_{t,l_t}^{-1} \left( \sum_{s \in \Psi_{t+1,l_t}} z_{s,l_t} \cdot \mathbf{a}_s \right)$.
7:     Define $C_{t,l_t} \leftarrow \{ \boldsymbol{\theta} \in \mathbb{R}^d : \|\boldsymbol{\theta} - \widehat{\boldsymbol{\theta}}_{t,l_t}\|^2_{\mathbf{V}_{t,l_t}^{-1}} \leq \beta_{t,l} \}$.
8:     Define $C_{t,l} \leftarrow C_{t-1,l}$ for all $l \in [L] \backslash \{l_t\}$.
9:     **Return** $\mathcal{C}_{t,l} \leftarrow \{ f_{\boldsymbol{\theta}} \in \mathcal{F} | \boldsymbol{\theta} \in C_{t,l} \}$ for all $l \in [L]$.
10: **end while**

---

### 6.3 REGRET BOUND OF ALGORITHM 1 WITH GLOC

With our new technical tool presented in the last subsection, we now show our final algorithm for the generalized linear bandit setting. We propose our algorithm in Algorithm 4. Generally speaking, Algorithm 4 is a *multi-level* version of the *generalized linear online-to-confidence-set conversion* (GLOC) algorithm proposed by Jun et al. (2017), equipped with FTRL.

As shown in Algorithm 4, we maintain $L$ FTRL online learners in parallel. Under our framework, a single learner only receives data with similar variances of noise. As a result, we can make use of the variance-dependent result shown in Theorem 6.5 to derive a tighter regret bound for generalized linear bandits with heteroscedastic noises.

**Theorem 6.8** (Regret bound for generalized linear bandits, informal). Suppose that Assumption 6.1 and 6.2 hold for the known reward function class $\mathcal{F}$. If we apply Algorithm 4 as a subroutine of Algorithm 1 (in line 9) and set $\beta_{t,l}$ to

$$1 + \frac{32 A^2 K^2 B^2}{\kappa^2} + \frac{26}{\kappa^2} R^2 \log^2(4t^2 L/\delta) + 12 \frac{2^{2(l+1)} \overline{\sigma}^2}{\kappa^2} d \log \left( 1 + \frac{t A \kappa^2}{4 d K^2} \right) + \lambda B^2$$

for all $t \in [T]$, $l \in [L]$, where $L = \lceil \log_2 R/\overline{\sigma} \rceil$, $\overline{\sigma} = R/\sqrt{d}$, then with probability $1 - 4\delta$, the regret of Algorithm 1 for the first $T$ rounds is bounded as follows:

$$\operatorname{Regret}(T) = \widetilde{O} \left( \frac{K}{\kappa} d \sqrt{J} + \frac{K}{\kappa} \left( K \cdot AB + R \right) \sqrt{dT} \right).$$

**Remark 6.9.** In the worst case, i.e. $\sigma_1 = \cdots = \sigma_T = R$, our result degraded to $\widetilde{O}(K R d \sqrt{T}/\kappa + K^2 A B \sqrt{dT}/\kappa)$, which still improves the $\widetilde{O} \left( K(KAB + R) d \sqrt{T}/\kappa \right)$ result provided by Jun et al. (2017).

**Remark 6.10.** Applying the regret bound in Corollary 5.4 in generalized linear bandits, we obtain a regret bound of $\widetilde{O} \left( K(d\sqrt{J} + d\sqrt{RT})/\kappa \right)$ for the case where $K \cdot A \cdot B = 1$ and $R = \Omega(1)$. Our bound here improved the general result when $R = o(d)$.

**Remark 6.11.** When restricted to the heteroscedastic linear bandits by setting $\kappa = K = 1$, our result becomes $\widetilde{O} \left( (R + A \cdot B) \sqrt{dT} + d\sqrt{J} \right)$, which is the same as the $\widetilde{O} \left( R\sqrt{dT} + d\sqrt{J} \right)$ regret in Zhou et al. (2021) when $R = \Omega(AB)$.

## 7 CONCLUSION AND FUTURE WORK

In this work we study heteroscedastic stochastic bandits problem for a general reward function class. We propose a multi-level regression framework ML$^2$ to deal with the heteroscedastic noises. Under three different settings with additional assumptions on the noise and the function class, we study the performance of ML$^2$ and propose corresponding variance-dependent regret bounds, which strictly improves previous algorithms for homoscedastic bandit setting. We leave to study the optimal regret bound of heteroscedastic stochastic bandits for a general reward function class for future work.

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

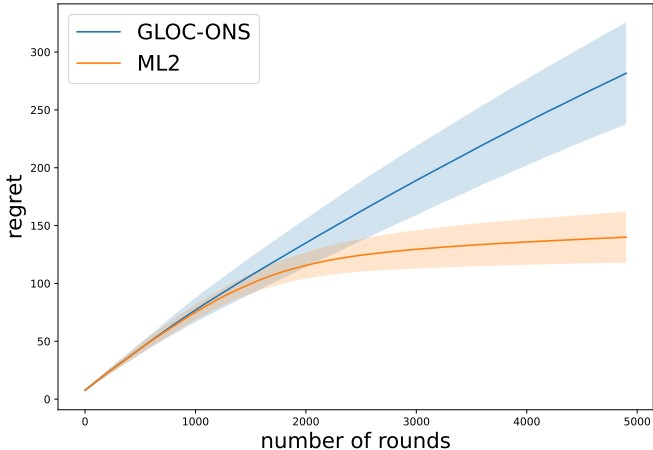

Figure 1: Cumulative regret comparison between ML$^2$ and GLOC on a synthetic data.

Zihan Zhang, Jiaqi Yang, Xiangyang Ji, and Simon S Du. Variance-aware confidence set: Variance-dependent bound for linear bandits and horizon-free bound for linear mixture mdp. *arXiv preprint arXiv:2101.12745*, 2021.

Dongruo Zhou, Quanquan Gu, and Csaba Szepesvari. Nearly minimax optimal reinforcement learning for linear mixture markov decision processes. In *Conference on Learning Theory*, pp. 4532–4576. PMLR, 2021.

## A   EXPERIMENTS

In this section, we conduct some experiments of the proposed ML$^2$+FTRL algorithm for generalized linear bandits. We compare our results with GLOC proposed by Jun et al. (2017). For each trial, the dimension of $\boldsymbol{\theta}^*$ is set to $d = 20$, and $\boldsymbol{\theta}^*$ is sampled uniformly from $[-\sqrt{1/d}, \sqrt{1/d}]^d$. At each round, the action set is of cardinality 100, where each action is uniformly sampled from $[-1, 1]^d$ and normalized to have a unit $\ell_2$ norm. The link function is set to be the sigmoid function $h(z) = \frac{1}{1+e^{-z}}$. The variance upper bound at each round is uniformly sampled from $[0, 0.75]$. To simulate the variance bounded condition considered in this paper, our reward is not sampled from Bernoulli distribution. Instead, at each round $t$, we sample a random variable $\epsilon'_t$ from Poisson distribution, i.e., $\widetilde{\epsilon}_t \sim \text{Pois}(\sigma_t^2)$ and $\epsilon'_t = \min\{\widetilde{\epsilon}_t, R\}$ where $R$ is set to 10. Our zero-mean noise $\epsilon_t$ is computed by $\epsilon_t = \epsilon'_t - \mathbb{E}[\epsilon'_t]$. With this construction, it is not hard to see that $\epsilon_t$ satisfies Assumption 3.5 with sub-Gaussian parameter $R$. We plot the cumulative regret of ML$^2$ and GLOC in Figure 1. We can see that by adapting the multi-level and variance-aware scheme, our ML$^2$ algorithm outperforms the previous GLOC algorithm by a large margin.

## B   PROOF SKECTCH OF THEOREM 5.3

*Proof Sketch.* We prove the result by showing the validity of the confidence sets and bounding the sum of single-step regret incurred by each variance level. **Step 1: Construction of confidence sets** We first show that the construction of $\mathcal{C}_{t,l}$ is big enough to contain $f^*$ with high probability. Recall the definition of $\mathcal{C}_{t,l}$ as follows

$$\mathcal{C}_{t,l} \leftarrow \left\{ f \in \mathcal{F} \Big| \sum_{s \in \Psi_{t,l}} \left[ f(a_s) - \widehat{f}_{t,l}(a_s) \right]^2 \leq \beta_{t,l} \right\} \tag{B.1}$$

Thus, it only suffices to show that $f^*$ satisfies the inequality in (B.1). According to our ERM sub-routine, we have

$$\sum_{s \in \Psi_{t,l}} (\widehat{f}_{t,l}(a_s) - f^*(a_s))^2 + 2 \sum_{s \in \Psi_{t,l}} \epsilon_s[f^*(a_s) - \widehat{f}_{t,l}(a_s)]$$

$$= \sum_{s \in \Psi_{t,l}} \left( \widehat{f}_{t,l}(a_s) - r_s \right)^2 - \sum_{s \in \Psi_{t,l}} (f^*(a_s) - r_s)^2 \leq 0,$$

where the inequality holds since $\widehat{f}_{t,l}$ is the function which minimizes the cumulative squared loss at level $l$, i.e., $\sum_{s \in \Psi_{t,l}} (f(a_s) - r_s)^2$. Therefore, to bound $\sum_{s \in \Psi_{t+1,l}} (\widehat{f}_{t+1,l}(a_s) - f^*(a_s))^2$, it suffices to bound the absolute value of $\sum_{s \in \Psi_{t,l}} \epsilon_s [f^*(a_s) - \widehat{f}_{t,l}(a_s)]$. Since $\widehat{f}$ suffers from the measurability issue, we use the concentration of self-normalized process with an $\alpha$-cover discretization argument to bound this term, which can finally show that $f^*$ satisfies (B.1). **Step 2: Regret decomposition** With the confidence sets corresponding to different variance levels constructed in Step 1, we decompose the final regret in the following way:

$$
\begin{aligned}
\text{Regret}(T) &\leq \sum_{l \in [L]} \sum_{t \in \Psi_{T+1,l}} (f^*(a_t^*) - f^*(a_t)) \\
&\leq \sum_{l \in [L]} \sum_{t \in \Psi_{T+1,l}} \left( \max_{f \in \mathcal{C}_{t,l}} f(a_t) - f^*(a_t) \right) \\
&\leq \sum_{l \in [L]} \sum_{t \in \Psi_{T+1,l}} w_{\mathcal{C}_{t,l}}(a_t) \quad\quad\quad (\text{B.2})
\end{aligned}
$$

where the second inequality holds due to the optimism principle for arm selection, the last one holds due to the definition of width $w_{\mathcal{C}_{t,l}}(a) := \max_{f \in \mathcal{C}_{t,l}} f(a) - \min_{f \in \mathcal{C}_{t,l}} f(a)$ and the fact that $f^* \in \mathcal{C}_{t,l}$ from Step 1. **Step 3: Bounding the sum of widths** From Step 2, the last step to bound the regret is to bound the summation of the width $w_{\mathcal{C}_{t,l}}(a_t)$. According to the definition of eluder dimension, we can further bound $w_{\mathcal{C}_{t,l}}(a_t)$ in terms of the eluder dimension of $\mathcal{F}$ and the value of $\beta$ (See Lemma E.3 for more details). According to the decomposition, we can first bound the regret at each level $l$ by a $\widetilde{O}(2^l \sqrt{|\Psi_{T+1,l}|})$ term (omitting the dependencies on $d$, $R$ and $\bar{\sigma}$) and then naturally bound the total regret by Lemma E.3 and the Cauchy Schwartz inequality. The full proof is given in Section F. □

## C EXAMPLE: VARIANCE-DEPENDENT REGRET BOUNDS FOR BERNOULLI BANDITS

In this section, we consider the following specific Bernoulli multi-armed bandits with general function approximation, where the observed reward $r_t$ incurred by action $a_t$ is subject to the following Bernoulli distribution:

$$
r_t = f^*(a_t) + \begin{cases} 1/f^*(a_t), & \text{with probability } f^*(a_t) \\ -1/(1 - f^*(a_t)), & \text{otherwise} \end{cases}, \xi \leq f^*(a_t) \leq 1/2. \quad (\text{C.1})
$$

It is easy to see that the noise $\epsilon_t$ satisfies $R = 1/f^*(a_t) \leq 1/\xi$ and $\sigma_t^2 = 1/f^*(a_t) + 1/(1 - f^*(a_t))$. As discussed in Remark 3.6, we can use estimators of variance in Algorithm 1 instead of the true variance at each round.

The following corollary is an illustration on how to apply Algorithm 1 when variance information is not accessible but can be estimated through past observation.

**Corollary C.1.** Let the same conditions as in Theorem 5.3 hold. Set $\alpha = T^{-2}, \bar{\sigma} = 1$. At each round, set $\sigma_t^2(a_t)$ to $\min\{1, \widehat{\sigma}^2(a_t) + 2/\xi^2 \cdot w_{\mathcal{C}_{t,l_t}}(a_t)\}$ where $\widehat{\sigma}^2(a_t) := 1/\widehat{f}_{t,l_t}(a_t) - 1/(1 - \widehat{f}_{t,l_t}(a_t))$. Then with probability at least $1 - \delta$, when $T$ is large enough, the regret of Algorithm 1 for the first $T$ rounds is bounded as $\text{Regret}(T) = \widetilde{O}(\sqrt{\dim_E \log N_\alpha \sum_{t=1}^{T} \text{Var}[r_t]} + \xi^{-3} \dim_E \log N_\alpha + \xi^{-1/2} \sqrt{\dim_E \log N_\alpha T})$.

*Proof.* We suppose the event described in Theorem 5.1 holds and the conditions of Theorem 5.3 hold without the given variance at each round.

At each round $t$, we can upper bound the variance of $\epsilon_t := r_t - f^*(a_t)$ by making use of $\mathcal{C}_{t,l}$ and $\widehat{f}_{t,l}$ returned by Algorithm 2. Specifically, we estimate $\text{Var}[r_t] = \sigma_t^2 = 1/f^*(a_t) + 1/(1 - f^*(a_t))$ as $1/\widehat{f}_{t,l_t}(a_t) - 1/(1 - \widehat{f}_{t,l_t}(a_t))$ and bound the gap between $\widehat{\sigma}_t^2$ and $\text{Var}[r_t]$:

$$
|\widehat{\sigma}_t^2(a_t) - \text{Var}[r_t]| \leq |1/\widehat{f}_{t,l_t}(a_t) - 1/f^*(a_t)| + |1/(1 - \widehat{f}_{t,l}(a_t)) - 1/(1 - f^*(a_t))|
$$

$$
\begin{aligned}
&\leq 2/\xi^2 \cdot |\widehat{f}_{t,l_t}(a_t) - f^*(a_t)| \\
&\leq 2/\xi^2 \cdot w_{\mathcal{C}_{t,l_t}}(a_t).
\end{aligned}
\tag{C.2}
$$

where the first inequality follows from the distribution of reward, and the last inequality holds according to Theorem 5.1.

Therefore, it is valid to replace $\sigma_t^2$ by $\min\{1, \widehat{\sigma}_t^2 + 2/\xi^2 \cdot w_{\mathcal{C}_{t,l_t}}(a_t)\}$ in Algorithm 1.

Similar to the proof of Theorem 5.3, we have

$$
\begin{aligned}
J &= \sum_{t=1}^T \min\{1, \widehat{\sigma}_t^2 + 2/\xi^2 \cdot w_{\mathcal{C}_{t,l_t}}(a_t)\} \\
&\leq \sum_{t=1}^T \min\{1, \mathrm{Var}[r_t] + 4/\xi^2 \cdot w_{\mathcal{C}_{t,l_t}}(a_t)\} \\
&\leq \sum_{t=1}^T \mathrm{Var}[r_t] + 4/\xi^2 \cdot \sum_{l \in [L]} \sum_{s \in \Psi_{T+1,l}} w_{\mathcal{C}_{s,l}}(a_s) \\
&\leq \sum_{t=1}^T \mathrm{Var}[r_t] + 4/\xi^2 \cdot L\left(1/T + d + 4\beta_{T,L}\sqrt{dT}\right)
\end{aligned}
$$

where the first equality follows from the definition of $J$, the first inequality follows from (C.2), the last inequality holds due to Lemma E.2.

Hence $J$ is of order $\widetilde{O}(\sum_{t=1}^T \mathrm{Var}[r_t] + 1/\xi^3 \cdot \sqrt{d \log N_\alpha T})$.

According to Theorem 5.3, the regret bound of Algorithm 1 for this Bernoulli bandit problem is of order

$$
\begin{aligned}
&\widetilde{O}\left(\sqrt{\dim_E \log N_\alpha \sum_{t=1}^T \mathrm{Var}[r_t]} + 1/\xi^{3/2} \cdot \dim_E^{3/4}(\log N_\alpha)^{3/4} T^{1/4} + \sqrt{\xi^{-1} \dim_E \log N_\alpha T}\right) \\
&\leq \widetilde{O}\left(\sqrt{\dim_E \log N_\alpha \sum_{t=1}^T \mathrm{Var}[r_t]} + \xi^{-3} \dim_E \log N_\alpha + \xi^{-1/2}\sqrt{\dim_E \log N_\alpha T}\right).
\end{aligned}
$$

This completes the proof. $\qquad\square$

# D  GAP-DEPENDENT REGRET BOUNDS

In this section, we provide gap-dependent regret bounds for Algorithm 1 with various subroutines.

Denote $\Delta_t$ as the smallest gap between the reward of an optimal action and the reward of a sub-optimal action:

$$
\Delta_t := \min_{a \in \mathcal{D}_t, a \notin \mathcal{D}_t^*} [f^*(a_t^*) - f^*(a)],
\tag{D.1}
$$

where $\mathcal{D}_t^* := \mathrm{argmax}_{a \in \mathcal{D}_t} f^*(a)$. Let $\Delta$ be the smallest gap in all the rounds: $\Delta := \min_{t \in [T]} \Delta_t$.

**Theorem D.1** (Gap-dependent regret bound for bandits with heteroscedastic sub-Gaussian noise). Suppose Assumption 3.4 and 4.1 hold and $|f^*(a)| \leq C$ for all $a \in \mathcal{A}$. Let $\sigma_{\max} = \max_{t \in [T]} \sigma_t$ and suppose $\sigma_{\max} > \overline{\sigma}$. If we apply Algorithm 2 as a subroutine of Algorithm 1 (in line 9) and set $\beta_{t,l}$ as the same value in Theorem 4.2, then with probability at least $1 - \delta$, the regret of Algorithm 1 for the first $T$ rounds is bounded as follows:

$$
\begin{aligned}
\mathrm{Regret}(T) &\leq \frac{L}{\Delta}\left(4\dim_E C^2 + 1/T\right) + 16\frac{LT\alpha C}{\Delta}\dim_E(\log T + 1) \\
&+ 128\frac{L}{\Delta}\sigma_{\max}^2 \log(2N_\alpha L/\delta)\dim_E(\log T + 1) + 32\frac{L}{\Delta}T\alpha\sigma_{\max}\sqrt{\log(8T^2L/\delta)}\dim_E(\log T + 1).
\end{aligned}
$$

**Corollary D.2.** Let the same conditions as in Theorem D.1 hold. Set $\alpha = T^{-2}$. Then with probability at least $1 - \delta$, when $T$ is large enough, the regret for the first $T$ rounds is bounded as follows:

$$
\mathrm{Regret}(T) = \widetilde{O}\left(\frac{\sigma_{\max}^2}{\Delta}\dim_E \log(\mathcal{N}(\mathcal{F}, T^{-2}, \|\cdot\|_\infty))\right).
$$

**Remark D.3.** Corollary D.2 immediately suggests an $\widetilde{O}(R^2 \dim_E \log(\mathcal{N}(\mathcal{F}, T^{-2}, \|\cdot\|_\infty))/\Delta)$ gap-dependent regret by the fact $\sigma_{\max} = O(R)$, which provides a novel instance-dependent bound for the original problem considered by Russo & Van Roy (2013). To our knowledge, this is the first result of its kind for the general bandit model.

**Remark D.4.** When restricted to linear contextual bandits with dimension $d$, our result reduces to $\widetilde{O}(\sigma_{\max}^2 d^2/\Delta)$, which matches the previous result derived in Abbasi-Yadkori et al. (2011).

**Theorem D.5** (Gap-dependent regret bound for bandits with heteroscedastic noise, informal). Suppose Assumption 3.4 holds and $|f^*(a)| \le 1$ for all $a \in \mathcal{A}$. Let $\sigma_{\max} = \max_{t \in [T]} \sigma_t$ (suppose $\sigma_{\max} > \bar{\sigma}$) and $d = \dim_E(\mathcal{F}, 1/T)$. If we apply Algorithm 2 as a subroutine of Algorithm 1 (in line 9) and set $\beta_{t,l}$ as the same value in Theorem 5.3, then with probability at least $1 - 2\delta$, the regret of Algorithm 1 for the first $T$ rounds is bounded as follows:

$$\text{Regret}(T) = \widetilde{O}\left(\frac{\sigma_{\max}^2}{\Delta} d \log N_\alpha + \frac{R}{\Delta} d \log N_\alpha\right)$$

if $T$ is large enough and we set $\alpha = T^{-2}$.

**Remark D.6.** Similar to Remark 5.5, compared with the regret in Corollary D.2, the regret in Theorem D.5 has an additional term that depends on $R$.

## E  PROOFS FROM SECTION 4

**Lemma E.1** (Proposition 3, Russo & Van Roy 2013). If $(\beta_t \ge 0 | t \in \mathbb{N})$ is a nondecreasing sequence and $\mathcal{F}_t := \left\{ f \in \mathcal{F} : \sum_{s=1}^{t-1} \left( \widehat{f}_t(a_s) - f(a_s) \right)^2 \le \beta_t^2 \right\}$, then

$$\sum_{t=1}^{T} \mathbb{1}(w_{\mathcal{F}_t}(a_t) > \epsilon) \le \left(\frac{4\beta_T^2}{\epsilon^2} + 1\right) \dim_E(\mathcal{F}, \epsilon)$$

for all $T \in \mathbb{N}$ and $\epsilon > 0$.

**Lemma E.2** (Lemma 2, Russo & Van Roy 2013). If $(\beta_t \ge 0 | t \in \mathbb{N})$ is a nondecreasing sequence and $\mathcal{F}_t := \left\{ f \in \mathcal{F} : \sum_{s=1}^{t-1} \left( \widehat{f}_t(a_s) - f(a_s) \right)^2 \le \beta_t^2 \right\}$, then

$$\sum_{t=1}^{T} w_{\mathcal{F}_t}(a_t) \le \frac{1}{T} + w_{\mathcal{F}}(\mathcal{A}) \cdot \dim_E(\mathcal{F}, T^{-2}) + 4\beta_T \sqrt{\dim_E(\mathcal{F}, T^{-2})T}$$

for all $T \in \mathbb{N}$.

Instead of using the previous approach that bounds the sum of widths, we take another approach to bound the sum of squared widths, which can further provide a novel gap-dependent result later.

**Lemma E.3** (Bounding the sum of the square of widths). If $(\beta_t \ge 0 | t \in \mathbb{N})$ is a nondecreasing sequence and $\mathcal{F}_t := \left\{ f \in \mathcal{F} : \sum_{s=1}^{t-1} \left( \widehat{f}_t(a_s) - f(a_s) \right)^2 \le \beta_t^2 \right\}$, then

$$\sum_{t=1}^{T} w_{\mathcal{F}_t}^2(a_t) \le \dim_E(\mathcal{F}, 1/\sqrt{T}) w_{\mathcal{F}}^2(\mathcal{A}) + 1 + 4\beta_T^2 \dim_E(\mathcal{F}, 1/\sqrt{T})(\log T + 1)$$

for all $T \in \mathbb{N}$.

*Proof.* Following a similar approach as Russo & Van Roy (2013), we reorder the set $\{w_{\mathcal{F}_t}(a_t)\}_{t \in [T]}$ to $\{w_t\}_{t \in [T]}$, such that $w_1 \ge w_2 \ge \cdots \ge w_T$.

Let $T' = \max\{t \in [T], w_t \ge \frac{1}{T}\}$. By Lemma E.1,

$$t \le \left(\frac{4\beta_T^2}{(w_t - \delta)^2} + 1\right) \dim_E(\mathcal{F}, w_t - \delta) \tag{E.1}$$

for any $\delta \in (0, \epsilon)$. Taking $\delta \to 0$, we have

$$w_t^2 \le \frac{4\beta_T^2 \dim_E(\mathcal{F}, w_t)}{t - \dim_E(\mathcal{F}, w_t)}. \tag{E.2}$$

Hence,

$$\sum_{t=1}^T w_{\mathcal{F}_t}^2(a_t) = \sum_{t=1}^T w_t^2 \tag{E.3}$$

$$\le \dim_E(\mathcal{F}, 1/T) w_{\mathcal{F}}^2(\mathcal{A}) + 1/T + \sum_{t=\dim_E(\mathcal{F}, 1/T)+1}^{T'} w_t^2$$

$$\le \dim_E(\mathcal{F}, 1/T) w_{\mathcal{F}}^2(\mathcal{A}) + 1/T + \sum_{t=\dim_E(\mathcal{F}, 1/T)+1}^{T'} \frac{4\beta_T^2 \dim_E(\mathcal{F}, 1/T)}{t - \dim_E(\mathcal{F}, 1/T)}$$

$$\le \dim_E(\mathcal{F}, 1/T) w_{\mathcal{F}}^2(\mathcal{A}) + 1/T + 4\beta_T^2 \dim_E(\mathcal{F}, 1/T)(\log T + 1),$$

where the first inequality holds due to $\sum_{t=T'+1}^T w_t^2 \le 1/T$ under our definition of $T'$, the second inequality follows from (E.2), the third inequality is derived by taking the integral. $\qquad\square$

With Assumption 4.1, we can directly apply the previous result on confidence set by replacing the sub-Gaussianity $\eta$ parameter by $2^{\ell+1}\overline{\sigma}$. Previous result by Russo & Van Roy (2013) achieved a confidence set of radius $\widetilde{O}(\sqrt{\eta^2 \log N_\alpha + \alpha t(C + \eta)})$. Ayoub et al. (2020) later provides a result of the same order with improvement in terms of smaller constants.

**Lemma E.4** (Theorem 5, Ayoub et al. 2020). Suppose that $|f^*(a)| \le C$ for all $a \in \mathcal{A}$. For any $\alpha > 0$, if we set

$$\beta_{t,l} = \left[ 8(2^{l+1} \cdot \overline{\sigma})^2 \log(2N_\alpha L/\delta) + 4t\alpha(C + \sqrt{(2^{l+1} \cdot \overline{\sigma})^2 \log(4t(t+1)L/\delta)}) \right]^{1/2}, \tag{E.4}$$

then with probability at least $1 - \delta$, for all $t \ge 1$, $l \in [L]$, $f^* \in \mathcal{C}_{t,l}$.

**Theorem E.5** (Restatement of Theorem 4.2). Suppose Assumptions 3.4 and 4.1 hold and $|f^*(a)| \le C$ for all $a \in \mathcal{A}$. For all $t \in [T], l \in [L]$ and $\delta \in (0,1), \alpha > 0, \overline{\sigma} > 0$, if we apply Algorithm 2 as a subroutine of Algorithm 1 (in line 9) and set $\beta_{t,l}$ as the square root of

$$8(2^{l+1} \cdot \overline{\sigma})^2 \log(2N_\alpha L/\delta) + 4t\alpha(C + \sqrt{(2^{l+1} \cdot \overline{\sigma})^2 \log(4t(t+1)L/\delta)}),$$

where $N_\alpha = \mathcal{N}(\mathcal{F}, \alpha, \|\cdot\|_\infty)$ and $L = \lceil \log_2 R/\overline{\sigma} \rceil$ (recall the definition of $L$ in Algorithm 1), then with probability at least $1 - \delta$, the regret for the first $T$ rounds is bounded as follows:

$$\text{Regret}(T) \le L + 2C \dim_E L + 8\sqrt{2L \dim_E(J + \overline{\sigma}^2 T) \log(2N_\alpha L/\delta)}$$

$$+ 4\sqrt{L \dim_E \alpha} \sqrt{C + 2R\sqrt{\log(4T(T+1)L/\delta)}T}.$$

*Proof.* For simplicity, let $d = \dim_E(\mathcal{F}, 1/T^2)$.

With probability at least $1 - \delta$, we have

$$\text{Regret}(T) = \sum_{t=1}^T (f^*(a_t^*) - f^*(a_t)) \tag{E.5}$$

$$= \sum_{l \in [L]} \sum_{t \in \Psi_{T+1,l}} (f^*(a_t^*) - f^*(a_t))$$

$$\le \sum_{l \in [L]} \sum_{t \in \Psi_{T+1,l}} \left( \max_{f \in \mathcal{C}_{t,l}} f(a_t) - f^*(a_t) \right)$$

$$\leq \sum_{l \in [L]} \left( 1 + 2C \cdot d + 4\beta_{T,l} \sqrt{d|\Psi_{T+1,l}|} \right)$$

$$\leq L + 2CdL + 2 \underbrace{\sqrt{L \sum_{l \in [L]} \beta_{T,l}^2 d |\Psi_{T+1,l}|}}_{I_0}, \tag{E.6}$$

where the first equality holds by the definition in (3.1), the second equality holds since $\Psi_{T+1,\cdot}$ forms a partition of $[T]$, the first inequality holds due to Lemma E.4, the second inequality follows from Lemma E.3, the third inequality is obtained by applying Cauchy-Schwarz inequality.

Then we continue to bound $I_0$,

$$I_0 = \sqrt{L \sum_{l \in [L]} \sum_{t \in \Psi_{T+1,l}} \beta_{T,l}^2 d}$$

$$\leq \sqrt{Ld} \sqrt{\sum_{l \in [L]} \sum_{t \in \Psi_{T+1,l}} 8(2^{l+1} \cdot \overline{\sigma})^2 \log(2N_\alpha L/\delta)}$$

$$+ \sqrt{Ld} \sqrt{\sum_{l \in [L]} \sum_{t \in \Psi_{T+1,l}} 4T\alpha(C + \sqrt{(4R^2 \cdot \log(4T(T+1)L/\delta))}}$$

$$\leq \sqrt{Ld} \left( \sqrt{\sum_{l \in [L]} \sum_{t \in \Psi_{T+1,l}} 8(2^{l+1} \cdot \overline{\sigma})^2 \log(2N_\alpha L/\delta)} + 2\sqrt{\alpha}T \sqrt{C + 2R\sqrt{\log(4T(T+1)L/\delta)}} \right)$$

$$\leq \sqrt{Ld} \left( \sqrt{\sum_{t=1}^{T} 32(\sigma_t^2 + \overline{\sigma}^2) \log(2N_\alpha L/\delta)} + 2\sqrt{\alpha}T \sqrt{C + 2R\sqrt{\log(4T(T+1)L/\delta)}} \right)$$

$$\leq \sqrt{32Ld(J + \overline{\sigma}^2 T) \log(2N_\alpha L/\delta)} + 2\sqrt{Ld\alpha} \sqrt{C + 2R\sqrt{\log(4T(T+1)L/\delta)}}T, \tag{E.7}$$

where the first inequality follows from the definition of $\beta_{T,l}$, the third inequality holds due to the fact that

$$\forall l \in [L], t \in \Psi_{T+1,l}, \quad 2^{l+1}\overline{\sigma} = 2 \cdot 2^l \overline{\sigma} \leq 2 \max\{\overline{\sigma}, \sigma_t\} = \sqrt{4 \max\{\overline{\sigma}^2, \sigma_t^2\}} \leq \sqrt{4(\overline{\sigma}^2 + \sigma_t^2)},$$

the fourth inequality follows from the definition of $J$.

Substituting (E.7) into (E.6), we obtain

$$\text{Regret}(T) \leq L + 2CdL + 8\sqrt{2Ld(J + \overline{\sigma}^2 T) \log(2N_\alpha L/\delta)}$$

$$+ 4\sqrt{Ld\alpha} \sqrt{C + 2R\sqrt{\log(4T(T+1)L/\delta)}}T,$$

which completes the proof. $\qquad \square$

**Theorem E.6** (Restatement of Theorem D.1). Suppose Assumptions 3.4 and 4.1 hold and $|f^*(a)| \leq C$ for all $a \in \mathcal{A}$. Let $\sigma_{\max} = \max_{t \in [T]} \sigma_t$. If we apply Algorithm 2 as a subroutine of Algorithm 1 (in line 9) and set $\beta_{t,l}$ as the same value in Theorem 4.2, then with probability at least $1 - \delta$, the regret of Algorithm 1 for the first $T$ rounds is bounded as follows:

$$\text{Regret}(T) \leq \frac{L}{\Delta} \left( 4 \dim_E C^2 + 1/T \right) + 16 \frac{LT\alpha C}{\Delta} \dim_E (\log T + 1)$$

$$+ 128 \frac{L}{\Delta} \sigma_{max}^2 \log(2N_\alpha L/\delta) \dim_E (\log T + 1)$$

$$+ 32 \frac{L}{\Delta} T\alpha\sigma_{max} \sqrt{\log(8T^2 L/\delta)} \dim_E (\log T + 1).$$

*Proof.* For simplicity, let $d = \dim_E(\mathcal{F}, 1/T^2)$.

Suppose the event described in Lemma E.4 holds. With probability at least $1 - \delta$,

$$
\begin{aligned}
&\text{Regret}(T) \\
&= \sum_{t=1}^{T} (f^*(a_t^*) - f^*(a_t)) \\
&= \sum_{l \in [L]} \sum_{t \in \Psi_{T+1,l}} (f^*(a_t^*) - f^*(a_t)) \\
&\leq \sum_{l \in [L]} \sum_{t \in \Psi_{T+1,l}} (f^*(a_t^*) - f^*(a_t))^2 / \Delta \\
&\leq \sum_{l \in [L]} \sum_{t \in \Psi_{T+1,l}} \left( \max_{f \in \mathcal{C}_{t,l}} f(a_t) - f^*(a_t) \right)^2 / \Delta \\
&\leq \frac{1}{\Delta} \sum_{l \in [L]} \sum_{t \in \Psi_{T+1,l}} w_{\mathcal{C}_{t,l}}^2 (\mathcal{D}_t) \\
&\leq \frac{1}{\Delta} \sum_{l \in [L]} \left( 4dC^2 + 1/T + 4\beta_{T,l}^2 d(\log T + 1) \right) \\
&\leq \frac{L}{\Delta} \cdot \left( 4dC^2 + 1/T \right) \\
&\quad + 4 \frac{L}{\Delta} d(\log T + 1) \left( 32\sigma_{max}^2 \log(2N_\alpha L / \delta) + 4T\alpha(C + 2\sqrt{\sigma_{max}^2 \log(4T(T+1)L/\delta)}) \right)
\end{aligned}
$$

where the first equality follows from the definition in (3.1), the second equality holds by the fact that $\Psi_{T+1,l}$ ($l \in [L]$) forms a partition of $[T]$, the first inequality holds due to the definition of $\Delta$ in Subsection 3.1, the second inequality follows from Lemma E.4, the fourth inequality holds due to Lemma E.3 and the last inequality is derived by directly substituting the value of $\beta_{T,l}$. □

## F PROOFS FROM SECTION 5

**Lemma F.1** (Freedman 1975)**.** Let $M, v > 0$ be fixed constants. Let $\{x_i\}_{i=1}^{n}$ be a stochastic process, $\{\mathcal{G}_i\}_i$ be a filtration so that for all $i \in [n]$, $x_i$ is $\mathcal{G}_i$-measurable, while most surely $\mathbb{E}[x_i | \mathcal{G}_{i-1}] = 0$, $|x_i| \leq M$ and

$$
\sum_{i=1}^{n} \mathbb{E}(x_i^2 | \mathcal{G}_i) \leq v.
$$

Then, for any $\delta > 0$, with probability $1 - \delta$, for all $t \in [n]$,

$$
\sum_{i=1}^{t} x_i \leq \sqrt{2v \log(2t^2/\delta)} + 2/3 \cdot M \log(2t^2/\delta).
$$

**Lemma F.2.** Suppose $a, b \geq 0$. If $x^2 \leq a + b \cdot x$, then $x^2 \leq 2b^2 + 2a$.

*Proof.* By solving the root of quadratic polynomial $q(x) := x^2 - b \cdot x - a$, we obtain $\max\{x_1, x_2\} = (b + \sqrt{b^2 + 4a})/2$. Hence, we have $x \leq (b + \sqrt{b^2 + 4a})/2$ provided that $q(x) \leq 0$. Then we further have

$$
x^2 \leq \frac{1}{4} \left( b + \sqrt{b^2 + 4a} \right)^2 \leq \frac{1}{4} \cdot 2 \left( b^2 + b^2 + 4a \right) \leq 2b^2 + 2a. \tag{F.1}
$$

□

**Theorem F.3** (Restatement of Theorem 5.1)**.** Suppose that $|f^*(a)| \leq C$ for all $a \in \mathcal{A}$. For any $\alpha > 0$ and $\delta \in (0, 1/2)$, if we set $\beta_{t,l}$ as the square root of

$$
12C\alpha t + 4\alpha \overline{R} t + \frac{8}{3} C \overline{R} \log(2N_\alpha t^2 / \delta) + 16 \cdot (2^{l+1} \overline{\sigma})^2 \log(2N_\alpha t^2 / \delta)
$$

where
$$\overline{R} = R\sqrt{2\log(4t^2/\delta)},$$

then $f^* \in \mathcal{C}_{t,l}$ for all $t$ with probability at least $1 - 2\delta$ for any fixed $l$.

*Proof.* By simple calculation, for all $f \in \mathcal{F}$ we have

$$\underbrace{\sum_{s\in\Psi_{t,l}} (f(a_s) - f^*(a_s))^2 + 2\sum_{s\in\Psi_{t,l}} \epsilon_s[f^*(a_s) - f(a_s)]}_{I(f)} = \sum_{s\in\Psi_{t,l}} (r_s - f(a_s))^2 - \sum_{s\in\Psi_{t,l}} (r_s - f^*(a_s))^2.$$

(F.2)

By sub-Gaussianity of $\epsilon_t$ we have

$$\mathbb{P}\left(\exists t \geq 1, \max_{1\leq s\leq t} |\epsilon_s| \geq R\sqrt{2\log(4t^2/\delta)}\right) \leq \sum_{s\geq 1} \mathbb{P}(|\epsilon_s| \geq R\sqrt{2\log(4s^2/\delta)}) \leq \sum_{s\geq 1} \delta/(2s^2) \leq \delta.$$

(F.3)

For simplicity, let event $\mathcal{E}_{\text{subG}} := \left\{\forall t \geq 1, \max_{1\leq s\leq t} |\epsilon_s| \leq R\sqrt{2\log(4t^2/\delta)}\right\}$.

Let $\mathcal{G}(\alpha) \subset \mathcal{F}$ be an $\alpha$-cover of $\mathcal{F}$ in $\|\cdot\|_\infty$.

From the definition of $\widehat{f}_{t,l}$, we have

$$\sum_{s\in\Psi_{t,l}} (\widehat{f}_{t,l}(a_s) - f^*(a_s))^2 + 2\sum_{s\in\Psi_{t,l}} \epsilon_s[f^*(a_s) - \widehat{f}_{t,l}(a_s)] = I(\widehat{f}_{t,l}) \leq 0. \qquad \text{(F.4)}$$

Let $g = \operatorname{argmin}_{\mathcal{G}(\alpha)} \|\widehat{f}_{t,l} - g\|_\infty$.

We then bound the gap $I(g) - I(\widehat{f}_{t,l})$ under event $\mathcal{E}_{subG}$,

$$I(g) - I(\widehat{f}_{t,l}) = \sum_{s\in\Psi_{t,l}} \left[(g(a_s) - f^*(a_s))^2 - (\widehat{f}_{t,l}(a_s) - f^*(a_s))^2\right] + 2\sum_{s\in\Psi_{t,l}} \epsilon_s[\widehat{f}_{t,l}(a_s) - g(a_s)]$$

$$\leq \sum_{s\in\Psi_{t,l}} (g(a_s) - \widehat{f}_{t,l}(a_s))(g(a_s) + \widehat{f}_{t,l}(a_s) - 2f^*(a_s)) + 2\sum_{s\in\Psi_{t,l}} \alpha R\sqrt{2\log(4t^2/\delta)}$$

$$\leq 4C\alpha t + 2\alpha R\sqrt{2\log(4t^2/\delta)}t. \qquad \text{(F.5)}$$

Fix an $f \in \mathcal{F}$. Applying Freedman's inequality (Lemma F.1), with probability at least $1 - \delta$, we have

$$\sum_{s\in\Psi_{t,l}} \epsilon_s \cdot \mathbb{1}(\mathcal{E}_{subG})[f^*(a_s) - f(a_s)] \geq -2/3R\sqrt{2\log(4t^2/\delta)}C\log(2t^2/\delta)$$

$$-\sqrt{2\cdot(2^{l+1}\overline{\sigma})^2 \sum_{s\in\Psi_{t,l}} (f(a_s) - f^*(a_s))^2 \log(2t^2/\delta)}.$$

(F.6)

for all $t \geq 1$.

Using a union bound on all the $f \in \mathcal{G}(\alpha)$ and $\mathcal{E}_{subG}$, we further obtain that

$$\sum_{s\in\Psi_{t,l}} \epsilon_t[f^*(a_s) - f(a_s)] \geq$$

$$-\frac{2}{3}R\sqrt{2\log(4t^2/\delta)}C\log(2N_\alpha t^2/\delta) - \sqrt{2(2^{l+1}\overline{\sigma})^2 \log(2N_\alpha t^2/\delta) \sum_{s\in\Psi_{t,l}} (f(a_s) - f^*(a_s))^2}$$

(F.7)

for all $f \in \mathcal{G}(\alpha)$ with probability at least $1 - 2\delta$.

Substituting (F.7) into the definition of $I(f)$, we have that for $g$, it holds for probability at least $1-2\delta$ that

$$4C\alpha t + 2\alpha R\sqrt{2\log(4t^2/\delta)}t \geq I(g) \tag{F.8}$$

$$\geq -\frac{4}{3}R\sqrt{2\log(4t^2/\delta)}C\log(2N_\alpha t^2/\delta) \tag{F.9}$$

$$-\sqrt{8\cdot(2^{l+1}\overline{\sigma})^2\log(2N_\alpha t^2/\delta)\sum_{s\in\Psi_{t,l}}(g(a_s)-f^*(a_s))^2} \tag{F.10}$$

$$+\sum_{s\in\Psi_{t,l}}(g(a_s)-f^*(a_s))^2 \tag{F.11}$$

where the first inequality is obtained by substituting (F.5) and (F.4) into the inequality below

$$I(g) \leq I(g) - I(\widehat{f}_{t,l}) + I(\widehat{f}_{t,l}),$$

the second inequality follows from the definition of $I(f)$ and (F.7).

Using Lemma F.2, we can deduce that

$$\sum_{s\in\Psi_{t,l}}(g(a_s)-f^*(a_s))^2 \leq 8C\alpha t + 4\alpha R\sqrt{2\log(4t^2/\delta)}t + \frac{8}{3}RC\sqrt{2\log(4t^2/\delta)}\log(2N_\alpha t^2/\delta)$$

$$+ 16\cdot(2^{l+1}\overline{\sigma})^2\log(2N_\alpha t^2/\delta).$$

Then we can complete the proof by bounding the gap between $\sum_{s\in\Psi_{t,l}}(g(a_s)-f^*(a_s))^2$ and $\sum_{s\in\Psi_{t,l}}(\widehat{f}_{t,l}(a_s)-f^*(a_s))^2$:

$$\sum_{s\in\Psi_{t,l}}(\widehat{f}_{t,l}(a_s)-f^*(a_s))^2 \leq \sum_{s\in\Psi_{t,l}}(g(a_s)-f^*(a_s))^2$$

$$+\left|\sum_{s\in\Psi_{t,l}}(g(a_s)-f^*(a_s))^2 - \sum_{s\in\Psi_{t,l}}(\widehat{f}_{t,l}(a_s)-f^*(a_s))^2\right|$$

$$\leq 12C\alpha t + 4\alpha R\sqrt{2\log(4t^2/\delta)}t + \frac{8}{3}RC\sqrt{2\log(4t^2/\delta)}\log(2N_\alpha t^2/\delta)$$

$$+ 16\cdot(2^{l+1}\overline{\sigma})^2\log(2N_\alpha t^2/\delta).$$

$\square$

**Theorem F.4** (Restatement of Theorem 5.3). Suppose Assumption 3.4 holds and $|f^*(a)| \leq 1$ for all $a \in \mathcal{A}$. For all $t \in [T], l \in [L]$ and $\delta \in (0,1), \alpha > 0, \overline{\sigma} > 0$, if we apply Algorithm 2 as a subroutine of Algorithm 1 (in line 9) and set $\beta_{t,l}$ as the square root of

$$12\alpha t + 4\alpha \overline{R}t + \frac{8}{3}\overline{R}\log(2N_\alpha t^2 L/\delta) + 16\cdot(2^{l+1}\overline{\sigma})^2\log(2N_\alpha t^2 L/\delta)$$

where $N_\alpha = \mathcal{N}(\mathcal{F}, \alpha, \|\cdot\|_\infty)$ and $\overline{R} = R\sqrt{2\log(4t^2 L/\delta)}$ (with a slight abuse of notation), then with probability at least $1 - 2\delta$, the regret for the first $T$ rounds is bounded as follows:

$$\text{Regret}(T) \leq 4\sqrt{L\dim_E(\log T + 1)\alpha}\sqrt{3 + \overline{R}T} + 2\sqrt{\frac{8}{3}L\dim_E(\log T + 1)\overline{R}\log(2N_\alpha t^2 L/\delta)T}$$

$$+ 16\sqrt{L\dim_E(\log T + 1)\log(2N_\alpha T^2 L/\delta)}\sqrt{J + T\overline{\sigma}^2} + \sqrt{L}\left(2\sqrt{\dim_E T} + 1\right).$$

*Proof.* For simplicity, let $d = \dim_E(\mathcal{F}, 1/T)$.

Based on Theorem 5.1, for any fixed $l$, we have $f^* \in \mathcal{C}_{t,l}$ with probability $1 - 2\delta/L$. Applying a union bound on all $l \in [L]$, we have $f^* \in \mathcal{C}_{t,l}$ for all $t, l$ with probability at least $1 - 2\delta$.

Then we further obtain, with probability at least $1 - 2\delta$,

$$\text{Regret}(T) = \sum_{t=1}^{T}(f^*(a_t^*) - f^*(a_t))$$

$$
\begin{aligned}
&= \sum_{l \in [L]} \sum_{t \in \Psi_{T+1,l}} (f^*(a_t^*) - f^*(a_t)) \\
&\leq \sum_{l \in [L]} \sum_{t \in \Psi_{T+1,l}} \left( \max_{f \in \mathcal{C}_{t,l}} f(a_t) - f^*(a_t) \right) \\
&= \sum_{l \in [L]} \sum_{t \in \Psi_{T+1,l}} \left( \max_{f \in \mathcal{C}_{t,l}} f(a_t) - f^*(a_t) \right) \sqrt{|\Psi_{T+1,l}|} \cdot \frac{1}{\sqrt{|\Psi_{T+1,l}|}} \\
&\leq \sqrt{L} \cdot \sqrt{\sum_{l \in [L]} |\Psi_{T+1,l}| \sum_{t \in \Psi_{T+1,l}} w_{\mathcal{C}_{t,l}}^2(\mathcal{D}_t)} \\
&\leq \sqrt{L} \cdot \sqrt{\sum_{l \in [L]} |\Psi_{T+1,l}| \left( 4d + 1/T + 4\beta_{T,l}^2 d(\log T + 1) \right)}
\end{aligned}
$$

where the first equality follows from the definition in (3.1), the second equality holds by the fact that $\Psi_{T+1,l}$ ($l \in [L]$) forms a partition of $[T]$, the second inequality follows from Cauchy-Schwarz inequality and Definition in (3.3), the third inequality follows from Lemma E.3.

Following the definition of $\beta_{t,l}$, we further calculate

$$
\begin{aligned}
\text{Regret}(T) &\leq \sqrt{L} \left( 2\sqrt{dT} + 1 \right) + \sqrt{L} \cdot \sqrt{\sum_{l \in [L]} \sum_{t \in \Psi_{T+1,l}} 64(2^{l+1}\overline{\sigma})^2 \log(2N_\alpha T^2 L/\delta) d(\log T + 1)} \\
&\quad + 2\sqrt{Ld(\log T + 1)T} \sqrt{12\alpha T + 4\alpha \overline{R} T + \frac{8}{3} \overline{R} \log(2N_\alpha T^2 L/\delta)} \\
&\leq \sqrt{L} \left( 2\sqrt{dT} + 1 \right) + 4\sqrt{Ld(\log T + 1)\alpha} \sqrt{3 + \overline{R}} T \\
&\quad + 2\sqrt{\frac{8}{3} Ld(\log T + 1)\overline{R} \log(2N_\alpha t^2 L/\delta) T} \\
&\quad + \sqrt{Ld(\log T + 1) \log(2N_\alpha T^2 L/\delta)} \sqrt{\sum_{t=1}^{T} 256(\sigma_t^2 + \overline{\sigma}^2)} \\
&\leq \sqrt{L} \left( 2\sqrt{dT} + 1 \right) + 4\sqrt{Ld(\log T + 1)\alpha} \sqrt{3 + \overline{R}} T \\
&\quad + 2\sqrt{\frac{8}{3} Ld(\log T + 1)\overline{R} \log(2N_\alpha t^2 L/\delta) T} \\
&\quad + 16\sqrt{Ld(\log T + 1) \log(2N_\alpha T^2 L/\delta)} \sqrt{J + T\overline{\sigma}^2},
\end{aligned}
$$

where the first inequality holds by the definition of $\beta_{t,l}$ and the fact that $\sqrt{a+b} \leq \sqrt{a} + \sqrt{b}$ for $a, b > 0$, the second inequality follows from the definition of $l_t$, the third inequality follows from the definition of $J$. $\square$

**Theorem F.5** (Formal version of Theorem D.5). Suppose Assumption 3.4 holds and $|f^*(a)| \leq 1$ for all $a \in \mathcal{A}$. Let $\sigma_{\max} = \max_{t \in [T]} \sigma_t$ and $d = \dim_E(\mathcal{F}, 1/T)$. If we apply Algorithm 2 as a subroutine of Algorithm 1 (in line 9) and set $\beta_{t,l}$ as the same value in Theorem 5.3, then with probability at least $1 - 2\delta$, the regret of Algorithm 1 for the first $T$ rounds is bounded as follows:

$$
\begin{aligned}
\text{Regret}(T) &\leq \frac{L}{\Delta}(4d + 1/T) + \frac{L\alpha T}{\Delta} d(\log T + 1)(12 + 4\overline{R}) + \frac{32}{3} \frac{L}{\Delta} \overline{R} d(\log T + 1) \log(2N_\alpha T^2 L/\delta) \\
&\quad + 256 \frac{L}{\Delta} \sigma_{\max}^2 d(\log T + 1) \log(2N_\alpha T^2 L/\delta).
\end{aligned}
$$

*Proof.* For simplicity, let $d = \dim_E(\mathcal{F}, 1/T)$.

Basically following the previous approach in Theorem D.1, with probability $1 - 2\delta$, we have

$$
\text{Regret}(T) = \sum_{t=1}^{T} (f^*(a_t^*) - f^*(a_t))
$$

$$
\begin{aligned}
&= \sum_{l \in [L]} \sum_{t \in \Psi_{T+1,l}} (f^*(a_t^*) - f^*(a_t)) \\
&\leq \sum_{l \in [L]} \sum_{t \in \Psi_{T+1,l}} (f^*(a_t^*) - f^*(a_t))^2 / \Delta \\
&\leq \sum_{l \in [L]} \sum_{t \in \Psi_{T+1,l}} \left( \max_{f \in \mathcal{C}_{t,l}} f(a_t) - f^*(a_t) \right)^2 / \Delta \\
&\leq \frac{1}{\Delta} \sum_{l \in [L]} \sum_{t \in \Psi_{T+1,l}} w_{\mathcal{C}_{t,l}}^2(\mathcal{D}_t) \\
&\leq \frac{1}{\Delta} \sum_{l \in [L]} \left( 4 + 1/T + 4\beta_{T,l}^2 d(\log T + 1) \right) \\
&\leq \frac{L}{\Delta} \cdot (4d + 1/T) \\
&\quad + 4 \frac{L}{\Delta} d(\log T + 1) \left( 12\alpha T + 4\alpha \overline{R} T + \frac{8}{3} \overline{R} \log(2 N_\alpha T^2 L / \delta) + 64 \sigma_{\max}^2 \log(2 N_\alpha T^2 L / \delta) \right) \\
&= \frac{L}{\Delta} (4d + 1/T) + \frac{L\alpha T}{\Delta} d(\log T + 1)(12 + 4\overline{R}) + \frac{32}{3} \frac{L}{\Delta} \overline{R} d(\log T + 1) \log(2 N_\alpha T^2 L / \delta) \\
&\quad + 256 \frac{L}{\Delta} \sigma_{\max}^2 d(\log T + 1) \log(2 N_\alpha T^2 L / \delta),
\end{aligned}
$$

where the first equality holds by the definition in (3.1), the second equality holds by the fact that $\Psi_{T+1,l}$ ($l \in [L]$) forms a partition of $[T]$, the first inequality follows from the definition of $\Delta$ in (D.1), the third inequality follows from Definition (3.3), the fourth inequality holds by Lemma E.3, the fifth inequality follows from the definition of $\beta_{T,l}$. $\qquad\square$

## G  PROOFS FROM SECTION 6

**Lemma G.1.** Let $\mathrm{reg}_t := \sum_{s=1}^t \ell(\mathbf{a}_s^\top \boldsymbol{\theta}, r_s) - \sum_{s=1}^t \ell(\mathbf{a}_s^\top \boldsymbol{\theta}^*, r_s)$. Following Algorithm 3, with probability at least $1 - \delta$,

$$
\sum_{s=1}^t (\mathbf{a}_s^\top (\boldsymbol{\theta}_s - \boldsymbol{\theta}^*))^2 \leq \frac{4}{\kappa} \mathrm{reg}_t + \frac{8R^2}{\kappa^2} \log(4t^2/\delta) \tag{G.1}
$$

for all $t \geq 1$. We denote the corresponding event by $\mathcal{E}_1$.

*Proof.*

$$
\begin{aligned}
\mathrm{reg}_t &= \sum_{s=1}^t \ell(\mathbf{a}_s^\top \boldsymbol{\theta}, r_s) - \sum_{s=1}^t \ell(\mathbf{a}_s^\top \boldsymbol{\theta}^*, r_s) \\
&= \sum_{s=1}^t \ell'(\mathbf{a}_s^\top \boldsymbol{\theta}^*, r_s) \mathbf{a}_s^\top (\boldsymbol{\theta}_s - \boldsymbol{\theta}^*) + \frac{\ell''(\xi_s, r_s)}{2} (\mathbf{a}_s^\top \boldsymbol{\theta}_s - \mathbf{a}_s^\top \boldsymbol{\theta}^*)^2 \\
&\geq -\sum_{s=1}^t \epsilon_s \mathbf{a}_s^\top (\boldsymbol{\theta}_s - \boldsymbol{\theta}^*) + \frac{\kappa}{2} (\mathbf{a}_s^\top \boldsymbol{\theta}_s - \mathbf{a}_s^\top \boldsymbol{\theta}^*)^2
\end{aligned}
$$

where the first equality follows from Definition 6.4, the second equality holds by Taylor series expansion and $\xi_s$ is a point between $\mathbf{a}_s^\top \boldsymbol{\theta}$ and $\mathbf{a}_s^\top \boldsymbol{\theta}^*$, the first inequality follows from Assumption 6.2 and Assumption 6.3.

Further we obtain

$$
\sum_{s=1}^t (\mathbf{a}_s^\top (\boldsymbol{\theta}_s - \boldsymbol{\theta}^*))^2 \leq \frac{2}{\kappa} \sum_{s=1}^t \epsilon_s \mathbf{a}_s^\top (\boldsymbol{\theta}_s - \boldsymbol{\theta}^*) + \frac{2}{\kappa} \mathrm{reg}_t
$$

$$\leq \frac{2}{\kappa} R \sqrt{2 \sum_{s=1}^{t} (\mathbf{a}_s^\top (\boldsymbol{\theta}_s - \boldsymbol{\theta}^*))^2 \log(2/\delta)} + \frac{2}{\kappa} \mathrm{reg}_t$$

with probability $1 - \delta$, where the second inequality follows from the sub-Gaussianity of $\epsilon_s$.

Applying Lemma F.2, we obtain that with probability at least $1 - \delta$,

$$\sum_{s=1}^{t} (\mathbf{a}_s^\top (\boldsymbol{\theta}_s - \boldsymbol{\theta}^*))^2 \leq \frac{4}{\kappa} \mathrm{reg}_t + \frac{8}{\kappa^2} R^2 \log(2/\delta).$$

Applying union bound on all $t \geq 1$, we have with probability at least $1 - \delta$,

$$\sum_{s=1}^{t} (\mathbf{a}_s^\top (\boldsymbol{\theta}_s - \boldsymbol{\theta}^*))^2 \leq \frac{4}{\kappa} \mathrm{reg}_t + \frac{8}{\kappa^2} R^2 \log(4t^2/\delta) \tag{G.2}$$

for all $t \geq 1$. $\qquad\qquad\square$

**Lemma G.2** (Lemma 11, Abbasi-Yadkori et al. 2011). *For any $\lambda > 0$ and sequence $\{\mathbf{x}_t\}_{t=1}^{T} \subset \mathbb{R}^d$ for $t \in \{0, 1, \cdots, T\}$, define $\mathbf{Z}_t = \lambda \mathbf{I} + \sum_{i=1}^{t} \mathbf{x}_i \mathbf{x}_i^\top$. Then, provided that $\|\mathbf{x}_t\|_2 \leq M$ for all $t \in [T]$, we have*

$$\sum_{t=1}^{T} \min\{1, \|\mathbf{x}_t\|_{\mathbf{Z}_{t-1}^{-1}}^2\} \leq 2d \log \frac{d\lambda + TM^2}{d\lambda}.$$

**Lemma G.3.** *For any $\lambda > 0$ and sequence $\{\mathbf{x}_t\}_{t=1}^{T} \subset \mathbb{R}^d$ for $t \in \{0, 1, \cdots, T\}$, define $\mathbf{Z}_t = \lambda \mathbf{I} + \sum_{i=1}^{t} \mathbf{x}_i \mathbf{x}_i^\top$. Then, provided that $\|\mathbf{x}_t\|_2 \leq M$ for all $t \in [T]$, we have*

$$\sum_{t=1}^{T} \|\mathbf{x}_t\|_{\mathbf{Z}_t^{-1}}^2 \leq 2d \log \frac{d\lambda + TM^2}{d\lambda}.$$

*Proof.* Applying matrix inversion lemma,

$$\begin{aligned}
\sum_{t=1}^{T} \|\mathbf{x}_t\|_{\mathbf{Z}_t^{-1}}^2 &= \sum_{t=1}^{T} \mathbf{x}_t^\top \mathbf{Z}_t^{-1} \mathbf{x}_t \\
&= \sum_{t=1}^{T} \mathbf{x}_t^\top \left( \mathbf{Z}_{t-1}^{-1} - \frac{\mathbf{Z}_{t-1}^{-1} \mathbf{x}_t \mathbf{x}_t^\top \mathbf{Z}_{t-1}^{-1}}{1 + \mathbf{x}_t^\top \mathbf{Z}_{t-1}^{-1} \mathbf{x}_t} \right) \mathbf{x}_t \\
&= \sum_{t=1}^{T} \frac{\|\mathbf{x}_t\|_{\mathbf{Z}_{t-1}^{-1}}^2}{1 + \|\mathbf{x}_t\|_{\mathbf{Z}_{t-1}^{-1}}^2} \\
&\leq \sum_{t=1}^{T} \min\{1, \|\mathbf{x}_t\|_{\mathbf{Z}_{t-1}^{-1}}^2\} \\
&\leq 2d \log \frac{d\lambda + TM^2}{d\lambda},
\end{aligned}$$

where the second equality follows from matrix inversion lemma, the second inequality holds by Lemma G.2. $\qquad\square$

**Lemma G.4.** *Let*

$$\boldsymbol{\Sigma}_t := \frac{4A^2 K^2}{\kappa} \mathbf{I} + \kappa \sum_{i=1}^{t} \mathbf{a}_i \cdot \mathbf{a}_i^\top,$$

$$\sigma_{\max} := \max_{t \geq 1} \sigma_t.$$

*Suppose $\overline{R}_t$ is an upper bound of $\max_{1 \leq s \leq t} |\epsilon_s|$. Then with probability at least $1 - \delta$, it holds simultaneously for all $t \geq 1$ that*

$$\sum_{s=1}^{t} \left( \epsilon_s^2 - \mathbb{E}[\epsilon_s^2] \right) \|\mathbf{a}_s\|_{\boldsymbol{\Sigma}_s^{-1}}^2 \leq \frac{\sigma_{\max} \overline{R}_t}{K} \sqrt{d \log \left( 1 + \frac{t A \kappa^2}{4dK^2} \log(2t^2/\delta) \right)} + \frac{2}{3\kappa} \left( \sigma_{\max}^2 + \overline{R}_t^2 \right) \log(2t^2/\delta).$$

*Proof.* To bound the sum of variance of each term, we caulculate

$$\sum_{s=1}^{t} \text{Var} \left[ \epsilon_s^2 - \mathbb{E}[\epsilon_s^2] \right] \|\mathbf{a}_s\|_{\mathbf{\Sigma}_s^{-1}}^4 \leq \sum_{s=1}^{t} \mathbb{E}[\epsilon_s^4] \|\mathbf{a}_s\|_{\mathbf{\Sigma}_s^{-1}}^4$$

$$\leq \sum_{s=1}^{t} \frac{\kappa}{4K^2} \mathbb{E}[\epsilon_s^2] \overline{R}_t^2 \|\mathbf{a}_s\|_{\mathbf{\Sigma}_s^{-1}}^2$$

$$\leq \sigma_{\max}^2 \frac{\overline{R}_t^2}{2K^2} d \log\left(1 + \frac{tA\kappa^2}{4dK^2}\right),$$

where the second inequality follows from the definition of $\mathbf{\Sigma}_t$, the third inequality holds by Lemma G.2.

Also note that

$$\max_{1 \leq s \leq t} \left(\epsilon_s^2 - \mathbb{E}[\epsilon_s^2]\right) \|\mathbf{a}_s\|_{\mathbf{\Sigma}_s^{-1}}^2 \leq \left(\sigma_{\max}^2 + \overline{R}_t^2\right) \frac{1}{\kappa}$$

since $\mathbf{\Sigma}_t \succeq \frac{4A^2K^2}{\kappa}\mathbf{I} + \kappa \mathbf{a}_s \cdot \mathbf{a}_s^\top$.

Then we apply Freedman's inequality, which gives for arbitrary $t \geq 1$,

$$\sum_{s=1}^{t} \left(\epsilon_s^2 - \mathbb{E}[\epsilon_s^2]\right) \|\mathbf{a}_s\|_{\mathbf{\Sigma}_s^{-1}}^2 \leq \frac{\sigma_{\max}\overline{R}_t}{K} \sqrt{d \log\left(1 + \frac{tA\kappa^2}{4dK^2}\right) \log(1/\delta)} + 2/3 \cdot \frac{1}{\kappa}\left(\sigma_{\max}^2 + \overline{R}_t^2\right) \log(1/\delta)$$

with probability at least $1 - \delta$.

Applying a union bound on all $t \geq 1$, we have with probability at least $1 - \delta$,

$$\sum_{s=1}^{t} \left(\epsilon_s^2 - \mathbb{E}[\epsilon_s^2]\right) \|\mathbf{a}_s\|_{\mathbf{\Sigma}_s^{-1}}^2 \leq \frac{\sigma_{\max}\overline{R}_t}{K} \sqrt{d \log\left(1 + \frac{tA\kappa^2}{4dK^2}\right) \log(2t^2/\delta)} + \frac{2}{3\kappa}\left(\sigma_{\max}^2 + \overline{R}_t^2\right) \log(2t^2/\delta)$$

(G.3)

since $\sum_{t \geq 1} \frac{\delta}{2t^2} \leq \delta$.

$\square$

**Theorem G.5** (Restatement of Theorem 6.5). If we set $\phi(\boldsymbol{\theta}) = \frac{2A^2K^2}{\kappa}\|\boldsymbol{\theta}\|_2^2$ and assume that all the data points fed into the algorithm are of noise variance bounded by $\sigma_{\max}^2$, then with probability at least $1 - 3\delta, \forall t \geq 1$, the regret of Algrithm 3 for the first $t$ rounds is bounded as follows:

$$\text{reg}_t \leq \frac{8A^2K^2B^2}{\kappa} + \frac{9}{2\kappa}R^2 \log^2(4t^2/\delta) + 3\frac{\sigma_{\max}^2}{\kappa}d\log\left(1 + \frac{tA\kappa^2}{4dK^2}\right).$$

*Proof.* For simplicity, let $\mathcal{L}_t(\boldsymbol{\theta}) = \sum_{s=1}^{t-1} \ell(\boldsymbol{\theta}^\top \mathbf{a}_s, r_s) + \phi(\boldsymbol{\theta})$ and $loss_t(\boldsymbol{\theta}) = \mathcal{L}_t(\boldsymbol{\theta}) - \phi(\boldsymbol{\theta})$.

Suppose event $\mathcal{E}_1$, event $\mathcal{E}_{subG} := \left\{\forall t \geq 1, \max_{1 \leq s \leq t} |\epsilon_s| \leq R\sqrt{2\log(4t^2/\delta)}\right\}$ and the event described in Lemma G.4 (denoted by $\mathcal{E}_2$) simultaneously hold in the following proof.

By sub-Gaussianity of $\epsilon_t$ we have

$$\mathbb{P}\left(\exists t \geq 1, \max_{1 \leq s \leq t} |\epsilon_s| \geq R\sqrt{2\log(4t^2/\delta)}\right) \leq \sum_{s \geq 1} \mathbb{P}(|\epsilon_s| \geq R\sqrt{2\log(4s^2/\delta)}) \leq \sum_{s \geq 1} \delta/(2s^2) \leq \delta.$$

(G.4)

Hence, $\mathbb{P}(\mathcal{E}_{subG}) \geq 1 - \delta$. Applying Lemma G.1 and Lemma G.4, we have that $\mathbb{P}(\mathcal{E}_{subG} \cap \mathcal{E}_1 \cap \mathcal{E}_2) \geq 1 - 3\delta$ by union bound.

From the update rule of Algorithm 3, we calculate

$$\sum_{s=1}^{t} \ell(\mathbf{a}_s^\top \boldsymbol{\theta}_s, r_s) - \sum_{s=1}^{t} \ell(\mathbf{a}_s^\top \boldsymbol{\theta}^*, r_s)$$

$$= \sum_{s=1}^{t} [loss_s(\boldsymbol{\theta}_s) - loss_{s+1}(\boldsymbol{\theta}_{s+1}) + \ell(\mathbf{a}_s^\top \boldsymbol{\theta}_s, r_s)] + loss_{t+1}(\boldsymbol{\theta}_{t+1}) - loss_{t+1}(\boldsymbol{\theta}^*)$$

$$= \sum_{s=1}^{t} [\mathcal{L}_{s+1}(\boldsymbol{\theta}_s) - \mathcal{L}_{s+1}(\boldsymbol{\theta}_{s+1})] - \phi(\boldsymbol{\theta}_1) + \phi(\boldsymbol{\theta}^*) + \mathcal{L}_{t+1}(\boldsymbol{\theta}_{t+1}) - \mathcal{L}_{t+1}(\boldsymbol{\theta}^*)$$

$$\leq 2 \max_{\boldsymbol{\theta} \in \Theta} |\phi(\boldsymbol{\theta})| + \sum_{s=1}^{t} \underbrace{[\mathcal{L}_{s+1}(\boldsymbol{\theta}_s) - \mathcal{L}_{s+1}(\boldsymbol{\theta}_{s+1})]}_{I_1}. \tag{G.5}$$

where the first equality follows from the definition of $loss$, the second equality holds by the definition of $\mathcal{L}$, the first inequality holds since $\mathcal{L}_{t+1}(\boldsymbol{\theta}_{t+1}) = \min_{\boldsymbol{\theta} \in \mathbb{R}^d} \mathcal{L}_{t+1}(\boldsymbol{\theta})$.

Then we continue to bound $I_1$.

$$I_1 = \sum_{s=1}^{t} [\mathcal{L}_{s+1}(\boldsymbol{\theta}_s) - \mathcal{L}_{s+1}(\boldsymbol{\theta}_{s+1})]$$

$$= \sum_{s=1}^{t} \left[ -\left\langle \frac{\partial \mathcal{L}_{s+1}}{\partial \boldsymbol{\theta}}(\boldsymbol{\theta}_s), \boldsymbol{\theta}_{s+1} - \boldsymbol{\theta}_s \right\rangle - (\boldsymbol{\theta}_{s+1} - \boldsymbol{\theta}_s)^\top \mathbf{H}_{s+1}(\boldsymbol{\theta}'_s)(\boldsymbol{\theta}_{s+1} - \boldsymbol{\theta}_s) \right]$$

$$= \sum_{s=1}^{t} \left[ -(h(\mathbf{a}_s^\top \boldsymbol{\theta}_s) - r_s) \langle \mathbf{a}_s, \boldsymbol{\theta}_{s+1} - \boldsymbol{\theta}_s \rangle - (\boldsymbol{\theta}_{s+1} - \boldsymbol{\theta}_s)^\top \mathbf{H}_{s+1}(\boldsymbol{\theta}'_s)(\boldsymbol{\theta}_{s+1} - \boldsymbol{\theta}_s) \right]$$

$$\leq \frac{1}{4} \sum_{s=1}^{t} (h(\mathbf{a}_s^\top \boldsymbol{\theta}_s) - r_s)^2 \|\mathbf{a}_s\|_{\mathbf{H}_{s+1}^{-1}(\boldsymbol{\theta}'_s)}^2$$

$$\leq \frac{1}{2} \sum_{s=1}^{t} (h(\mathbf{a}_s^\top \boldsymbol{\theta}_s) - h(\mathbf{a}_s^\top \boldsymbol{\theta}^*))^2 \|\mathbf{a}_s\|_{\mathbf{H}_{s+1}^{-1}(\boldsymbol{\theta}'_s)}^2 + \frac{1}{2} \sum_{s=1}^{t} \epsilon_s^2 \|\mathbf{a}_s\|_{\mathbf{H}_{s+1}^{-1}(\boldsymbol{\theta}'_s)}^2$$

$$\leq \frac{1}{2} K^2 \underbrace{\sum_{s=1}^{t} (\mathbf{a}_s^\top \boldsymbol{\theta}_s - \mathbf{a}_s^\top \boldsymbol{\theta}^*)^2 \|\mathbf{a}_s\|_{\mathbf{H}_{s+1}^{-1}(\boldsymbol{\theta}'_s)}^2}_{I_2} + \frac{1}{2} \underbrace{\sum_{s=1}^{t} \epsilon_s^2 \|\mathbf{a}_s\|_{\mathbf{H}_{s+1}^{-1}(\boldsymbol{\theta}'_s)}^2}_{I_3} \tag{G.6}$$

where the second equality holds due to Taylor Expansion, (Let $\mathbf{H}$ be the Hessian matrix and $\boldsymbol{\theta}'_s \in \mathbb{R}^d$), the third equality follows from the fact that $\frac{\partial \mathcal{L}_s}{\partial \boldsymbol{\theta}}(\boldsymbol{\theta}_s) = 0$ and $\frac{\partial \mathcal{L}_{s+1}}{\partial \boldsymbol{\theta}}(\boldsymbol{\theta}_s) = \frac{\partial \mathcal{L}_s}{\partial \boldsymbol{\theta}}(\boldsymbol{\theta}_s) + \frac{\partial \ell(\boldsymbol{\theta}^\top \mathbf{a}_s, r_s)}{\partial \boldsymbol{\theta}}(\boldsymbol{\theta}_s)$ the first inequality is obtained by solving the quadratic function with respect to $\boldsymbol{\theta}_{s+1} - \boldsymbol{\theta}_s$, the second inequality follows from $(a + b)^2 \leq 2a^2 + 2b^2$.

From the definition of $\mathbf{H}$, we calculate

$$\mathbf{H}_{s+1}(\boldsymbol{\theta}) = \frac{\partial}{\partial \boldsymbol{\theta}} [\nabla_{\boldsymbol{\theta}} \mathcal{L}_{s+1}] \tag{G.7}$$

$$= \frac{\partial}{\partial \boldsymbol{\theta}} \left[ \sum_{i=1}^{s} (h(\mathbf{a}_i^\top \boldsymbol{\theta}) - r_i) \cdot \mathbf{a}_i + \nabla_{\boldsymbol{\theta}} \phi \right] \tag{G.8}$$

$$\succeq \frac{4A^2 K^2}{\kappa} \mathbf{I} + \kappa \sum_{i=1}^{s} \mathbf{a}_i \cdot \mathbf{a}_i^\top = \boldsymbol{\Sigma}_s. \tag{G.9}$$

From Lemma G.1,

$$I_2 \leq \left[ \frac{4}{\kappa} \text{reg}_t + \frac{8R^2}{\kappa^2} \log(4t^2/\delta) \right] \frac{\kappa}{4K^2}$$

$$\leq \frac{1}{K^2} \text{reg}_t + \frac{2R^2}{\kappa K^2} \log(4t^2/\delta). \tag{G.10}$$

We bound $I_3$ by decomposing it into its expected value and a zero-mean term.

$$I_3 \leq \sum_{s=1}^{t} \sigma_s^2 \|\mathbf{a}_s\|_{\mathbf{H}_{s+1}^{-1}(\boldsymbol{\theta}'_s)}^2 + \sum_{s=1}^{t} (\epsilon_s^2 - \mathbb{E}[\epsilon_s^2]) \|\mathbf{a}_s\|_{\mathbf{H}_{s+1}^{-1}(\boldsymbol{\theta}'_s)}^2$$

$$\leq \sum_{s=1}^{t} \sigma_s^2 \|\mathbf{a}_s\|_{\mathbf{\Sigma}_s^{-1}}^2 + \sum_{s=1}^{t} \left(\epsilon_s^2 - \mathbb{E}[\epsilon_s^2]\right) \|\mathbf{a}_s\|_{\mathbf{\Sigma}_s^{-1}}^2$$

$$\leq 2 \frac{\sigma_{\max}^2}{\kappa} d \log\left(1 + \frac{tA\kappa^2}{4dK^2}\right) + \underbrace{\sum_{s=1}^{t} \left(\epsilon_s^2 - \mathbb{E}[\epsilon_s^2]\right) \|\mathbf{a}_s\|_{\mathbf{\Sigma}_s^{-1}}^2}_{I_4}, \tag{G.11}$$

where the second inequality follows from (G.9), the third inequality holds by Lemma G.3.

Substituting (G.11) and (G.10) into (G.6), we have

$$I_1 \leq \frac{1}{2}\mathrm{reg}_t + \frac{R^2}{\kappa}\log(4t^2/\delta) + \frac{\sigma_{\max}^2}{\kappa}d\log\left(1 + \frac{tA\kappa^2}{4dK^2}\right) + \frac{1}{2}I_4. \tag{G.12}$$

Applying Lemma G.4 to bound $I_4$, we calculate

$$I_4 \leq \frac{\sigma_{\max}\overline{R}_t}{K}\sqrt{d\log\left(1 + \frac{tA\kappa^2}{4dK^2}\right)\log(2t^2/\delta)} + \frac{2}{3\kappa}\left(\sigma_{\max}^2 + \overline{R}_t^2\right)\log(2t^2/\delta)$$

$$\leq \frac{\sigma_{\max}R}{K}\sqrt{2d\log\left(1 + \frac{tA\kappa^2}{4dK^2}\right)\log(4t^2/\delta)} + 2/3 \cdot \left(\sigma_{\max}^2 + 2R^2\log(4t^2/\delta)\right)\frac{1}{\kappa}\log(4t^2/\delta)$$

$$\leq \frac{\sigma_{\max}R}{K}\sqrt{2d\log\left(1 + \frac{tA\kappa^2}{4dK^2}\right)\log(4t^2/\delta)} + \frac{2R^2}{\kappa}\log^2(4t^2/\delta), \tag{G.13}$$

where the second inequality holds due to event $\mathcal{E}_{subG}$, the third inequality follows from $\sigma_{\max} \leq R$.

Substituting (G.13) into (G.12), we have

$$I_1 \leq \frac{1}{2}\mathrm{reg}_t + \frac{R^2}{\kappa}\log(4t^2/\delta) + \frac{\sigma_{\max}^2}{\kappa}d\log\left(1 + \frac{tA\kappa^2}{4dK^2}\right)$$

$$+ \frac{\sigma_{\max}R}{2K}\sqrt{2d\log\left(1 + \frac{tA\kappa^2}{4dK^2}\right)\log(4t^2/\delta)} + \frac{R^2}{\kappa}\log^2(4t^2/\delta)$$

$$\leq \frac{1}{2}\mathrm{reg}_t + \frac{2R^2}{\kappa}\log^2(4t^2/\delta) + \frac{\sigma_{\max}^2}{\kappa}d\log\left(1 + \frac{tA\kappa^2}{4dK^2}\right)$$

$$+ \frac{1}{2\kappa}\sigma_{\max}^2 d\log\left(1 + \frac{tA\kappa^2}{4dK^2}\right) + \frac{1}{4\kappa}R^2\log^2(4t^2/\delta)$$

$$= \frac{1}{2}\mathrm{reg}_t + \frac{9}{4\kappa}R^2\log^2(4t^2/\delta) + \frac{3}{2}\frac{\sigma_{\max}^2}{\kappa}d\log\left(1 + \frac{tA\kappa^2}{4dK^2}\right). \tag{G.14}$$

Substituting the upper bound of $I_1$ above to (G.5),

$$\mathrm{reg}_t \leq 2\max_{\boldsymbol{\theta}\in\Theta}|\phi(\boldsymbol{\theta})| + \frac{1}{2}\mathrm{reg}_t + \frac{9}{4\kappa}R^2\log^2(4t^2/\delta) + \frac{3}{2}\frac{\sigma_{\max}^2}{\kappa}d\log\left(1 + \frac{tA\kappa^2}{4dK^2}\right)$$

$$\leq \frac{8A^2K^2B^2}{\kappa} + \frac{9}{2\kappa}R^2\log^2(4t^2/\delta) + 3\frac{\sigma_{\max}^2}{\kappa}d\log\left(1 + \frac{tA\kappa^2}{4dK^2}\right), \tag{G.15}$$

from which we can further complete the proof by the arbitrariness of $t$.

$$\square$$

In the following lemma, we formally introduce the conversion from online learning regret to confidence set in our setting. For simplicity in analysis, we omit the level subscript and suppose all the data is fed into the same level.

**Lemma G.6.** Suppose we feed loss function $\{\ell_s(\boldsymbol{\theta})\}_{s=1}^{t}$ into a single online learner $\mathcal{B}$. Assume that $\mathcal{B}$ has an online learning (OL) regret bound $\overline{\mathrm{reg}}_t$: $\forall t \geq 1$,

$$\sum_{s=1}^{t} \ell_s(\boldsymbol{\theta}_s) - \ell_s(\boldsymbol{\theta}^*) \leq \overline{\mathrm{reg}}_t. \tag{G.16}$$

Define $\mathbf{X}_t$ as the design matrix consisting of $\mathbf{a}_1, \cdots, \mathbf{a}_t$, $\mathbf{z}_t = [z_1, \cdots, z_t]$. Then, with probability at least $1 - 4\delta$,

$$\forall t \geq 1, \|\boldsymbol{\theta}^* - \widehat{\boldsymbol{\theta}}_t\|_{\overline{\mathbf{V}}_t}^2 \leq 1 + \frac{4}{\kappa}\overline{\mathrm{reg}}_t + \frac{8R^2}{\kappa^2}\log\left(4t^2/\delta\right) + \lambda B^2 - \left(\|\mathbf{z}_t\|_2^2 - \widehat{\boldsymbol{\theta}}_t^\top\mathbf{X}_t^\top\mathbf{z}_t\right). \quad \text{(G.17)}$$

*Proof.* With Lemma G.1, we can prove this lemma by following nearly the same proof for Theorem 1 in Jun et al. (2017). (We can set $\beta_t'$ in their proof to be $1 + \frac{4}{\kappa}\overline{\mathrm{reg}}_t + \frac{8R^2}{\kappa^2}\log\left(4t^2/\delta\right)$ according to Lemma G.1. ) □

**Lemma G.7.** *For all $t$, with $\mathbf{z}_t$ and $\mathbf{X}_t$ defined as in Lemma G.8, we have*

$$\|\mathbf{z}_t\|_2^2 - \widehat{\boldsymbol{\theta}}_t^\top\mathbf{X}_t^\top\mathbf{z}_t \geq 0.$$

*Proof.* After ridge regression, $\widehat{\boldsymbol{\theta}}_t = \overline{\mathbf{V}}_t^{-1}\mathbf{X}_t^\top\mathbf{z}_t$ where $\overline{\mathbf{V}}_t := \lambda\mathbf{I} + \mathbf{X}_t^\top\mathbf{X}_t$.

Then we have

$$\begin{aligned}
\|\mathbf{z}_t\|_2^2 - \widehat{\boldsymbol{\theta}}_t\mathbf{X}_t^\top\mathbf{z}_t &= \|\mathbf{z}_t\|_2^2 - \left(\overline{\mathbf{V}}_t^{-1}\mathbf{X}_t^\top\mathbf{z}_t\right)^\top\mathbf{X}_t^\top\mathbf{z}_t \\
&= \|\mathbf{z}_t\|_2^2 - \mathbf{z}_t^\top\mathbf{X}_t\overline{\mathbf{V}}_t^{-1}\mathbf{X}_t^\top\mathbf{z}_t \quad \text{(G.18)}
\end{aligned}$$

We consider

$$\begin{bmatrix} \lambda\mathbf{I} + \mathbf{X}_t^\top\mathbf{X}_t & \mathbf{X}_t^\top \\ \mathbf{X}_t & \mathbf{I} \end{bmatrix} \succeq [\mathbf{X}_t \quad \mathbf{I}]^\top [\mathbf{X}_t \quad \mathbf{I}] \succeq \mathbf{0}. \quad \text{(G.19)}$$

From Schur complement theorem, we have

$$\mathbf{I} \succeq (\mathbf{I} + \mathbf{X}_t^\top\mathbf{X}_t)^{-1}\mathbf{X}_t^\top = \mathbf{X}_t\overline{\mathbf{V}}_t^{-1}\mathbf{X}_t^\top. \quad \text{(G.20)}$$

Then we can complete the proof by substituting (G.20) into (G.18). □

**Lemma G.8** (Variance-dependent confidence set for generalized linear bandits)**.** Suppose that Assumption 6.1, 6.2, 6.3 hold. For any $\delta \in (0, 1/4)$, if we set

$$\beta_{t,l} := 1 + \frac{32A^2K^2B^2}{\kappa^2} + \frac{26}{\kappa^2}R^2\log^2(4t^2L/\delta) + 12\frac{2^{2(l+1)}\overline{\sigma}^2}{\kappa^2}d\log\left(1 + \frac{tA\kappa^2}{4dK^2}\right) + \lambda B^2, \quad \text{(G.21)}$$

then with probability at least $1 - 4\delta$, we have $\boldsymbol{\theta}^* \in C_{t,l}$ for all $t \geq 1, l \in [L]$.

*Proof.* For any $l \in [L]$, with probability at least $1 - \frac{\delta}{L}$,

$$\|\boldsymbol{\theta}^* - \widehat{\boldsymbol{\theta}}_{t,l}\|_{\overline{\mathbf{V}}_{t,l}}^2 \leq 1 + \frac{4}{\kappa}\overline{\mathrm{reg}}_{t,l} + \frac{8R^2}{\kappa^2}\log\left(4t^2L/\delta\right) + \lambda B^2,$$

making use of Lemma G.8 and Lemma G.7.

From Theorem 6.5, with probability at least $1 - 3\frac{\delta}{L}$, we can set

$$\overline{\mathrm{reg}}_{t,l} := \frac{8A^2K^2B^2}{\kappa} + \frac{9}{2\kappa}R^2\log^2(4t^2L/\delta) + 3\frac{2^{2(l+1)}\overline{\sigma}^2}{\kappa}d\log\left(1 + \frac{tA\kappa^2}{4dK^2}\right). \quad \text{(G.22)}$$

By Union Bound, with probability $1 - 4\delta$, for all $l \in [L]$,

$$\begin{aligned}
\|\boldsymbol{\theta}^* - \widehat{\boldsymbol{\theta}}_{t,l}\|_{\overline{\mathbf{V}}_{t,l}}^2 &\leq 1 + \frac{32A^2K^2B^2}{\kappa^2} + \frac{18}{\kappa^2}R^2\log^2(4t^2L/\delta) + 12\frac{2^{2(l+1)}\overline{\sigma}^2}{\kappa^2}d\log\left(1 + \frac{tA\kappa^2}{4dK^2}\right) \\
&\quad + \frac{8R^2}{\kappa^2}\log\left(4t^2L/\delta\right) + \lambda B^2 \\
&= 1 + \frac{32A^2K^2B^2}{\kappa^2} + \frac{26}{\kappa^2}R^2\log^2(4t^2L/\delta) + 12\frac{2^{2(l+1)}\overline{\sigma}^2}{\kappa^2}d\log\left(1 + \frac{tA\kappa^2}{4dK^2}\right) + \lambda B^2.
\end{aligned}$$

□

**Theorem G.9.** Suppose that Assumptions 6.1 and 6.2 hold for the known reward function class $\mathcal{F}$. If we apply Algorithm 4 as a subroutine of Algorithm 1 (in line 9) and set $\beta_{t,l}$ to

$$1 + \frac{32A^2K^2B^2}{\kappa^2} + \frac{26}{\kappa^2}R^2\log^2(4t^2L/\delta) + 12\frac{2^{2(l+1)}\overline{\sigma}^2}{\kappa^2}d\log\left(1 + \frac{tA\kappa^2}{4dK^2}\right) + \lambda B^2$$

for all $t \in [T]$, $l \in [L]$, $\overline{\sigma} = R/\sqrt{d}$, then with probability $1 - 4\delta$, the regret of Algorithm 1 for the first $T$ rounds is bounded as follows:

$$\text{Regret}(T) = \widetilde{O}\left(\frac{K}{\kappa}d\sqrt{J} + \frac{K}{\kappa}\left(K \cdot AB + R\right)\sqrt{dT}\right).$$

*Proof.*

$$
\begin{aligned}
\text{Regret}(T) &= \sum_{t \in [T]} h(\mathbf{x}_{t,*}^\top \boldsymbol{\theta}^*) - h(\mathbf{x}_t^\top \boldsymbol{\theta}^*) \\
&\leq K \sum_{t \in [T]} (\mathbf{x}_{t,*}^\top \boldsymbol{\theta}^* - \mathbf{x}_t^\top \boldsymbol{\theta}^*) \\
&\leq K \sum_{t \in [T]} \left( \max_{\boldsymbol{\theta} \in \cap_{l \in [L]} C_{t-1,l}} \mathbf{x}_t^\top \boldsymbol{\theta} - \mathbf{x}_t^\top \boldsymbol{\theta}^* \right) \\
&\leq K \sum_{l \in [L]} \sum_{t \in \Psi_{T+1,l}} \left( \max_{\boldsymbol{\theta} \in C_{t,l}} \mathbf{x}_t^\top \boldsymbol{\theta} - \mathbf{x}_t^\top \boldsymbol{\theta}^* \right) \\
&\leq K \sum_{l \in [L]} \sum_{t \in \Psi_{T+1,l}} \min\left( 2AB, \|\mathbf{x}_t\|_{\overline{\mathbf{V}}_{t-1,l}^{-1}} \max_{\boldsymbol{\theta}_1, \boldsymbol{\theta}_2 \in C_{t-1,l}} \|\boldsymbol{\theta}_1 - \boldsymbol{\theta}_2\|_{\overline{\mathbf{V}}_{t-1,l}} \right) \\
&\leq 2K \sum_{l \in [L]} \sum_{t \in \Psi_{T+1,l}} \min\left( \beta_{t,l}^{1/2} \|\mathbf{x}_t\|_{\overline{\mathbf{V}}_{t-1,l}^{-1}}, AB \right), \quad\quad\quad\quad (\text{G.23})
\end{aligned}
$$

where the first equality holds by the definition in (3.1), the first inequality follows from Assumption 6.2, the second inequality holds by Lemma G.8.

For an arbitrary $l \in [L]$, let $\mathcal{I}_1(l) := \left\{ t \in \Psi_{T+1,l} \,\middle|\, \|\mathbf{x}_t\|_{\overline{\mathbf{V}}_{t-1,l}^{-1}} \leq 1 \right\}$ and $\mathcal{I}_2(l) := \Psi_{T+1,l} \backslash \mathcal{I}_1(l)$.

$$
\begin{aligned}
\sum_{t \in \mathcal{I}_1} \min\left( \beta_{t,l}^{1/2} \|\mathbf{x}_t\|_{\overline{\mathbf{V}}_{t-1,l}^{-1}}, AB \right) &\leq \beta_{T,l}^{1/2} \sum_{t \in \mathcal{I}_1} \min(1, \|\mathbf{x}_t\|_{\overline{\mathbf{V}}_{t-1,l}^{-1}}) \\
&\leq \sqrt{\beta_{T,l} |\Psi_{T+1,l}| \sum_{t \in \mathcal{I}_1} [\min(1, \|\mathbf{x}_t\|_{\overline{\mathbf{V}}_{t-1,l}^{-1}})]^2} \\
&\leq \sqrt{2\beta_{T,l} |\Psi_{T+1,l}| d\log\frac{d\lambda + TA^2}{d\lambda}}, \quad\quad (\text{G.24})
\end{aligned}
$$

where the second inequality holds by Cauchy-Schwartz inequality, the third inequality follows from Lemma G.2.

Similarly, we calculate

$$
\begin{aligned}
\sum_{t \in \mathcal{I}_2} \min\left( \beta_{t,l}^{1/2} \|\mathbf{x}_t\|_{\overline{\mathbf{V}}_{t-1,l}^{-1}}, AB \right) &\leq AB \sum_{t \in \mathcal{I}_2} \min(1, \|\mathbf{x}_t\|_{\overline{\mathbf{V}}_{t-1,l}^{-1}}) \\
&\leq AB \sqrt{|\Psi_{T+1,l}| \sum_{t \in \mathcal{I}_1} [\min(1, \|\mathbf{x}_t\|_{\overline{\mathbf{V}}_{t-1,l}^{-1}})]^2} \\
&\leq AB \sqrt{2|\Psi_{T+1,l}| d\log\frac{d\lambda + TA^2}{d\lambda}}. \quad\quad (\text{G.25})
\end{aligned}
$$

Substituting (G.24) and (G.25) into (G.23), we obtain

$$
\begin{aligned}
\text{Regret} &\leq 2K \sum_{l \in [L]} (AB + \sqrt{\beta_{T,l}}) \sqrt{2|\Psi_{T+1,l}| d \log \frac{d\lambda + TA^2}{d\lambda}} \\
&\leq 2K \sqrt{L \sum_{l \in [L]} 2(AB + \sqrt{\beta_{T,l}})^2 d |\Psi_{T+1,l}| \log \frac{d\lambda + TA^2}{d\lambda}} \\
&\leq 4K\sqrt{L} \sqrt{A^2 B^2 dT \log \frac{d\lambda + TA^2}{d\lambda} + \sum_{l \in [L]} \beta_{T,l} d |\Psi_{T+1,l}| \log \frac{d\lambda + TA^2}{d\lambda}} \\
&\leq 4K\sqrt{L} \left( 1 + \frac{4\sqrt{2}AKB}{\kappa} + \frac{6}{\kappa} R \log(4T^2 L/\delta) + \sqrt{\lambda}B + AB \right) \sqrt{dT \log \frac{d\lambda + TA^2}{d\lambda}} \\
&\quad + 4K\sqrt{L} \sqrt{\sum_{l \in [L]} 12 \frac{d^2}{\kappa^2} (2^{l+1}\overline{\sigma})^2 |\Psi_{T+1,l}| \log \frac{d\lambda + TA^2}{d\lambda} \log \left( 1 + \frac{tA\kappa^2}{4dK^2} \right)} \\
&\leq 4K\sqrt{L} \left( 1 + \frac{4\sqrt{2}AKB}{\kappa} + \frac{6}{\kappa} R \log(4T^2 L/\delta) + \sqrt{\lambda}B + AB \right) \sqrt{dT \log \frac{d\lambda + TA^2}{d\lambda}} \\
&\quad + 4K \sqrt{L \log \frac{d\lambda + TA^2}{d\lambda} \log \left( 1 + \frac{tA\kappa^2}{4dK^2} \right)} \sqrt{\sum_{l \in [L]} \sum_{t \in \Psi_{T+1,l}} 12 \frac{d^2}{\kappa^2} (2^{l+1}\overline{\sigma})^2} \\
&\leq 4K\sqrt{L} \left( 1 + \frac{4\sqrt{2}AKB}{\kappa} + \frac{6}{\kappa} R \log(4T^2 L/\delta) + \sqrt{\lambda}B + AB \right) \sqrt{dT \log \frac{d\lambda + TA^2}{d\lambda}} \\
&\quad + 4K \sqrt{L \log \frac{d\lambda + TA^2}{d\lambda} \log \left( 1 + \frac{tA\kappa^2}{4dK^2} \right)} \sqrt{\sum_{t \in [T]} 48 \frac{d^2}{\kappa^2} (\sigma_t^2 + \overline{\sigma}^2)} \\
&\leq 4K\sqrt{L} \left( 1 + \frac{4\sqrt{2}AKB}{\kappa} + \frac{6}{\kappa} R \log(4T^2 L/\delta) + \sqrt{\lambda}B + AB \right) \sqrt{dT \log \frac{d\lambda + TA^2}{d\lambda}} \\
&\quad + 24 \frac{K}{\kappa} d \sqrt{L \log \frac{d\lambda + TA^2}{d\lambda} \log \left( 1 + \frac{tA\kappa^2}{4dK^2} \right)} \sqrt{J + \overline{\sigma}^2 T}
\end{aligned}
$$

where the second inequality holds due to Cauchy-Schwarz inequality, the fourth inequality follows from the definition of $\beta_{t,l}$, the sixth inequality holds since $l_t$ in Algorithm 1 satisfies $2^{l_t}\overline{\sigma} \leq \max\{\sigma_t, \overline{\sigma}\}$, the last inequality follows from the definition of $J$. $\qquad\square$

