# OpenReview forum: "Bandit Learning with General Function Classes: Heteroscedastic Noise and Variance-dependent Regret Bounds"
_ICLR.cc/2023/Conference — Submitted to ICLR 2023_

### Official Review · Reviewer_pkxU · 2022-10-22

**Confidence:** 4
**Correctness:** 4
**Technical Novelty And Significance:** 2
**Empirical Novelty And Significance:** Not applicable
**Recommendation:** 5

**Clarity, Quality, Novelty And Reproducibility:**

My main concern with this paper is in terms of novelty, as I am unable to pinpoint the main novel contribution of this work.
- In terms of algorithmic framework, the proposed $ML^2$ algorithm is essentially a minor variant of the sup-family of algorithms, i.e., Sup-Lin-UCB (Auer, 2002), Sup-Kernel-UCB (Valko et al, 2013). The only difference is that the authors use the $\sigma_t^2$ provided by the environment instead of say, given by the GP model in Sup-Kernel-UCB. Hence, from the algorithmic viewpoint, $ML^2$ offers limited novelty.
- In terms of analysis, most of the analysis is based upon combining existing results in related work. The only difference from other works is the use of $\sigma_t$, instead of a global bound $R$, which from my understanding of the appendix does not seem to offer any major technical hurdle. Moreover, since the authors also do not point a novelty in terms of analysis in the main text, it reinforces my earlier point.
- In terms of regret bounds: From the "main contributions" section in the introduction, it seems that most of the contribution is in terms of improved regret bounds that have a variance-dependent term. However, it is not fair to compare the bounds obtained in this work directly with classical results in homoscedastic noise. This is because under this setup the author assume access to additional information about $\sigma_t$, which is not assumed in the homoscedastic version. From what I understand, the variance-dependent improvement in the regret bounds comes due to availability of this additional information. Furthermore, the access to this additional information and its use in algorithm takes away most of the challenge in algorithm design in the heteroscedastic setup.

So overall, unless I am missing out on something major, I don't see the contribution of this paper at this stage to be novel enough to warrant a publication.

EDIT:
After discussion with authors, I am increasing the score a bit. However, as mentioned in the comment to them, I still don't think the novelty in analysis is strong enough to warrant a publication.

**Strength And Weaknesses:**

Strengths:
- The paper considers an important problem with real applications and has proposed several algorithm to tackle the challenges of heteroscedastic noise that offer improved performance over existing ones.

Weaknesses:
In additional to the comments about novelty in the next section, I have following questions/comments.
- The algorithm assumes availability of $\sigma_t$ from the environment which is different from estimating it. Can the authors support the feasibility of such an assumption by perhaps an actual application?
- In Sec. 5. it says, "In this section, we consider the original setting introduced in Section 3 without Assumption 4.1.", where Assumption 4.1 is the assumption sub-Gaussianity on the noise. However, in proof of theorem 5.1, sub-Gaussianity of noise plays an important role and the result of this theorem is used throughout the section. Can the authors please explain this inconsistency?
- While I understand that the main contributions of this work are theoretical in nature, I feel it is important to provide empirical evidence, especially while submitting it to a conference like ICLR. I am also interested in seeing the practical performance of these algorithms because of their similarity to Sup-family of algorithms (see next section) which are known to perform poorly in practice. Having some simulations might provide better insight into the algorithm performance and its ability to leverage heteroscedastic noise.
- As a minor comment, I think the exposition would benefit by adding some more intuition about the algorithms while describing them.
- Typos: provaly -> provably (pg 2, second paragraph), variaces -> variances (pg 5, last paragraph before Sec. 4).



**Summary Of The Paper:**

The paper considers the problem of general functions from a general function class under a bandit feedback model. Specifically, this work considers the setup of heteroscedastic noise where in the noise varies with different actions, possibly depending on the action taken by the learner. Under this scenario, the authors proposes a new algorithm that can leverage the heteroscedasticity of the noise and incurs a regret bound dependent on it, allowing it to take advantage of low noise regimes. The authors also propose a variant for the generalized linear bandit model and establish its regret performance.

**Summary Of The Review:**

As mentioned above, I am not convinced about the contribution being novel enough, which is why I am giving the paper a score of 3.

---

> ### Author Response · Authors · 2022-11-17
> **Response to Reviewer pkxU (Part 2)**
>
> **Q5**: In terms of analysis, most of the analysis is based upon combining existing results in related work. The only difference from other works is the use of $\sigma_t$, instead of a global bound $R$, which from my understanding of the appendix does not seem to offer any major technical hurdle.
>
> **A5**: We respectfully disagree with your comment on this point. $\sigma_t$ is the standard variance of noise while $R$ is the variance proxy of the sub-Gaussian noise. Although at the high level, the strategy of ML2 can be summarized as using $\sigma_t$ to ‘replace’ $R$, there still exist several technical difficulties when designing a proper scheme to utilize $\sigma_t$ and deriving a tighter regret bound. We summarize the main difficulties w.r.t. each section as follows.
>
> In Sec. 4, the main difficulty is dealing with the heterogeneous variances. For the linear contextual bandits case, we can adopt a weighted regression framework proposed by Zhou et al. (2021) which naturally leads to a dependence on the summation of variances. However, such a strategy cannot be applied to the general nonlinear function class since the function class is not closed under a linear mapping (we have demonstrated it in ‘Existing approach’ paragraph in Sec. 3). The ML2 framework is then proposed as a solution to this issue.
> In Sec. 6, the main difficulty is removing the additional $d\sqrt{RT}$ term in the regret proposed in Thm. 4.2 under the generalized linear bandit setting. To achieve this goal, again we cannot simply replace $R$ with $\sigma_t$ for the existing results in Jun et al. (2017) or make a simple replacement from Thm. 4.2. We first propose a refined regret bound for online regression in Thm. 6.5, which improves the existing result in Jun et al. (2017). Then we run the GLOC framework with a multi-level modification (note that algorithm design is novel since it originates from ML2!) and obtain a better regret in Thm. 6.8. We believe these results and analysis are new and novel in the generalized linear bandit setting.
>
>
> ----
>
> **Q6**: ’the variance-dependent improvement in the regret bounds comes due to the availability of this additional information. Furthermore, the access to this additional information and its use in algorithm takes away most of the challenge in algorithm design in the heteroscedastic setup.’
>
> **A6**: We admit that the known variance assumption is stronger and has not been utilized in previous work. However, even given the variance information, it is still challenging to design an algorithm that can utilize the variances condition efficiently. Note that the only previous algorithms which can utilize the variance information are weighted OFUL in Zhou et al. (2021) and VOFUL in Zhang et al. (2021). However, neither of them can be applied to the general nonlinear function class: weighted OFUL cannot deal with the general function class which is not closed under linear mapping, while VOFUL is computationally intractable due to the implicitly constructed confidence sets. Thus, ML2 is the first computationally tractable algorithm for heteroscedastic bandits with general function approximation, and we believe ML2 indeed solves a challenge existing in the bandits with general function class literature.
>
> In addition, as we mentioned above, we have provided an example where the variance is unknown but can be estimated on the fly in section B. We hope this will remove your concern.

---

> ### Author Response · Authors · 2022-11-17
> **Response to Reviewer pkxU (Part 1)**
>
> Thank you for your insightful comments!
>
> **Q1**: The algorithm assumes availability of $\sigma_t$ from the environment which is different from estimating it. Can the authors support the feasibility of such an assumption by perhaps an actual application?
>
> **A1**: Yes, we have provided an example where the variance is unknown but can be estimated on the fly in section B. We hope this will remove your concern.
>
> ----
>
> **Q2**: In Sec. 5. it says, "In this section, we consider the original setting introduced in Section 3 without Assumption 4.1.", where Assumption 4.1 is the assumption of sub-Gaussianity on the noise. However, in the proof of theorem 5.1, the sub-Gaussianity of noise plays an important role and the result of this theorem is used throughout the section.
>
> **A2**: We are sorry for the confusion. In Section 5, we still assume that Assumption 3.5 holds, which is a strictly weaker assumption than Assumption 4.1. We have made it clearer in the revision.
>
> ----
>
> **Q3**: Having some simulations might provide better insight into the algorithm's performance and its ability to leverage heteroscedastic noise.
>
> **A3**: Thank you for your suggestion! While the main contribution of our paper is more on the theoretical side, we have added an experiment to validate the main theoretical results in Appendix A. In detail, we have conducted experiments on the proposed ML$^2$+FTRL algorithm for generalized linear bandits. We compare our algorithm with GLOC proposed by Jun et al. (2017). For each trial, the dimension of $\theta^*$ is set to $d = 20$, and $\theta^*$ is sampled uniformly from $[-\sqrt{1 / d}, \sqrt{1 / d}]^d$. At each round, the action set is of cardinality $100$, where each action is uniformly sampled from $[-1, 1]^d$ and normalized to a unit 2-norm vector.  The link function is set to be the sigmoid function $h(z) = \frac{1}{1 + e^{-z}}$. The variance upper bound at each round is uniformly sampled from $[0, 0.75]$. To simulate the variance-bounded condition considered in this paper, our reward is not sampled from Bernoulli distribution. Instead, at each round $t$, we sample a random variable $\epsilon_t'$ from Poisson distribution, i.e., $\tilde{\epsilon}_t \sim Pois(\sigma_t^2)$ and $\epsilon_t' = \min\{\tilde{\epsilon}_t, R\}$ where $R$ is set to 10. And our zero-mean noise $\epsilon_t$ is computed by $\epsilon_t = \epsilon_t' - \mathbb{E}[\epsilon_t']$. With this construction, it is not hard to see that $\epsilon_t$ satisfies Assumption 3.5 with sub-Gaussian parameter $R$. From Figure 1, we can see that by adapting the multi-level and variance-aware structure, our ML$^2$ algorithm outperforms the GLOC algorithm by a large margin.
>
> We hope this experiment can help remove your concern.
>
> ----
>
>
> **Q4**: 'In terms of algorithmic framework, the proposed ML2 algorithm is essentially a minor variant of the sup-family of algorithms, i.e., Sup-Lin-UCB (Auer, 2002), Sup-Kernel-UCB (Valko et al, 2013). The only difference is that the authors use the $\sigma_t^2$ provided by the environment instead of saying, given by the GP model in Sup-Kernel-UCB. Hence, from the algorithmic viewpoint, ML2 offers limited novelty.'
>
> **A4**: We respectfully disagree with your comment. Our ML2 algorithm is very different from Sup-family algorithms in the following aspects. From the algorithm design perspective, sup-algorithm groups the incoming contexts into different levels based on their exploration term $\\| x \\|\_{A^{-1}}$, while ML2 groups the contexts based on the variances of their corresponding rewards. The difference in the algorithm design is due to the different goals: Sup-algorithms aim to reduce the dependence on dimension $d$ from $d$ to $\sqrt{d}$, while ML2 aims to reduce the dependence on the variance from $R\sqrt{T}$ to $\sqrt{\sum_{t=1}^T \sigma_t^2}$. Thus, although ML2 shares a similar multi-level structure as Sup-algorithms, they are essentially very different.

---

> > ### Comment · Reviewer_pkxU · 2022-11-19
> > **Reply to authors**
> >
> > Thanks for the detailed response.
> >
> > 1. I am sorry I can't find section B. Are you referring to the appendix or something else? If you are referring to section B in Appendix, the current version does not have a description of such an example.
> >
> > 3. Thanks for the adding the simulation. I think it helps makes a stronger case for your work. As an aside, can you check if the performance stays similar if $\sigma_t$ is sampled from $[1,2]$ instead of $[0, 0.75]$. I suspect that might not be the case. (I don't expect a response for this particular experiment, but I feel it might help put through my point more clearly).
> >
> > 4. $\|x\|_{A^{-1}}$ corresponds to the uncertainty in the estimate which is exactly what $\sigma_t$ represents in your framework. So just because it is called something different, I don't think it really changes the algorithm much. I agree the ultimate objective is different, but given the similarity to existing algorithms, the multi-level framework does not seem to be a significantly novel contribution.
> >
> > 5. If I understand correctly, the main analytical contribution is towards improving the bound from $O(d \sqrt{RT})$ to $O(R \sqrt{dT})$?

---

> > > ### Author Response · Authors · 2022-11-21
> > > **Re: Reply to authors**
> > >
> > > Thank you for your feedback and further questions.
> > >
> > >
> > > **Q1**: Are you referring to the appendix or something else? If you are referring to section B in Appendix, the current version does not have a description of such an example.
> > >
> > > **A1**: We are sorry for the typo. We wanted to refer to Section C in the appendix, where an example with Bernoulli reward is given along with the corresponding algorithm and analysis.
> > >
> > > ----
> > >
> > > **Q2**: Thanks for adding the simulation. I think it helps makes a stronger case for your work. As an aside, can you check if the performance stays similar if $\sigma_t$ is sampled from [1,2] instead of [0,0.75]. I suspect that might not be the case.
> > >
> > > **A2**: Thank you for your suggestion. We have conducted additional experiments on the new setting suggested by you, where $\sigma_t$  is sampled from [1,2]. The experiments are summarized in the following table, which summarizes the cumulative regret for GLOC (Jun et al. 2017) and our algorithm ML2. It can be seen that similar to the setting of $\sigma_t \sim [0,0.75]$, our algorithm ML2 outperforms GLOC by a large margin. This again corroborates our theoretical result.
> > > | Algorithm     | t = 5000               | t = 7500 |t = 10000   |
> > > |--------------|-----------------------|--------------------|-------------------|
> > > | ML2 | 300.49   | 426.71 | 549.21 |
> > > | GLOC | 324.02    | 477.23 | 625.73 |
> > >
> > > ----
> > >
> > >
> > > **Q3**: $\\|x\\|_{A^{-1}}$ represents the uncertainty in the estimate which is exactly $\sigma_t$ in your work…. algorithms are similar
> > >
> > > **A3**: This is not true. $\\|x\\|\_{A^{-1}}$ and $\sigma_t$ are fundamentally different quantities. First, $\\|x\\|\_{A^{-1}}$ represents the uncertainty of the \emph{estimated} reward, while $\sigma_t$ represents the uncertainty of the \emph{true} reward. Second, $\\|x\\|\_{A^{-1}}$ depends on the algorithm while $\sigma_t$ does not depend on the algorithm. For a bandit algorithm with good exploration, the uncertainty of the \emph{estimated} rewards $\\|x\\|\_{A^{-1}}$  may decrease as the number of time rounds increases, while $\sigma_t$ cannot be decreased no matter how long the algorithm runs (it only depends on the environment). So they’re totally different things.
> > >
> > > For the algorithm design, we would also like to point out that the original Sup-family algorithms are more complex than our ML2 algorithm. An important characteristic is that the number of levels in Sup-family algorithms depends on the number of total rounds (i.e., $L = O(\log{T})$ according to Chu et al. (2011)). In contrast, the number of levels of ML2 is independent of the number of rounds (where $L=O(\log (R/\bar\sigma)$). That suggests that Sup-family algorithms will lead to more levels than ML2 especially when $T$ is large. Thus, even from the algorithm design perspective, ML2 is quite different from Sup-family algorithms.
> > >
> > > ----
> > >
> > > **Q4**:  If I understand correctly, the main analytical contribution is towards improving the bound from $O(d\sqrt{RT})$ to $O(R\sqrt{dT})$?
> > >
> > > **A4**:  Your understanding of the main contribution regarding our Section 6 is correct. We proved a new theorem (Theorem 6.5), which gives a tighter regret for FTRL, and is of independent interest. The improved regret for FTRL also leds to the improved regret for our ML2 algorithm. Besides, we also made significant contributions in Section 5. In detail, we derive a new variance dependent confidence set in Theorem 5.1, which is novel since $\sigma$ here stands for standard variance instead of the sub-Gaussian parameter. Prior to our work, a variance-dependent upper confidence bound for linear contextual bandits is given by Zhou et al. (2021). Their analysis highly relies on the algebraic structure of the $R^d$ space and the Bernstein Inequality for vector-valued martingales. As a result, their analysis cannot be applied for general nonlinear reward function or generalized linear bandit setting.
> > > To obtain Theorem 5.1, our key technique is to apply an $\alpha$-net covering of the reward function class and then bound $\sum \epsilon_t (f(x_t) - f^*(x_t))$ using Freedman’s inequality, which is a totally different analysis compared to the analysis used in Zhou et al. (2021).
> > >
> > > —-----------------------------------------------

---

> > > > ### Comment · Reviewer_pkxU · 2022-12-12
> > > > **Reply to authors**
> > > >
> > > > Thank you for your reply and I apologize for a late response.
> > > >
> > > > I still don't think there is any fundamental difference between the Sup-family and ML framework. The only differences are superficial. With that said, I might have misunderstood some theoretical contributions. I am increasing the score a bit, but I still don't think the theoretical result in itself can carry this paper through for a publication.

---

> > > ### Author Response · Authors · 2022-11-29
> > > **Follow up with Reviewer pkxU**
> > >
> > > Dear Reviewer pkxU,
> > >
> > > Thanks for your reply!
> > >
> > > Since the deadline for the author-reviewer discussion phase is fast approaching, we would like to follow up with you to see if you have any further questions. In our response, we have addressed all your additional questions. Specifically, we add additional experiments to remove your further concern about our algorithm. In response to your concern about the novelty of our algorithm, we also explain the differences between the two frameworks on how to partition past observations.
> > >
> > > We are looking forward to your feedback.
> > >
> > > Best,
> > > Authors of Paper 3544

---

### Official Review · Reviewer_npEC · 2022-10-24

**Confidence:** 3
**Clarity, Quality, Novelty And Reproducibility:** The paper is well written.
**Correctness:** 4
**Technical Novelty And Significance:** 2
**Empirical Novelty And Significance:** Not applicable
**Recommendation:** 5

**Strength And Weaknesses:**

Strength: Their results are solid. Their proposed "multi-level" framework has some novelty.

Weakness:  The known variance assumptions largely reduce the difficulty of the whole problem. With this info and given the previous VOFUL algorithm which also considers the multi-level variance, this idea seems not very surprising to me. Furthermore, its generalized linear function results also seem a bit incremental. (Their first improvement that change R to $\sigma_\text{max}$ seems very direct given the variance is known. Their further improvement on d is not clear to me how significant it is...)

**Summary Of The Paper:**

This work considers the bandits' problem with general function classes and heteroscedastic noise. Under the assumption that the variance of the noise is simultaneously given with a reward after each round, they give the first variance-aware regret bound in terms of Eluder dimension. Their main technical innovation is to propose a multi-level learning framework that partition the historical data into several layers according to the given variance and applies base bandits algorithms on that. Besides the result on general function classes, they further give results on generalized linear functions under this "multi-level" framework. By using the generalized linear online-to-confidence-set conversion (GLOC) algorithm proposed by Jun et al. (2017), equipped with FTRL, they show that the results under such function classes can be further improved by $\widetilde{O}\left(\sqrt{\operatorname{dim}_E \log N_\alpha R T}\right)$

**Summary Of The Review:**

The results are solid and clear. My only concern is their techniques are a bit incremental and their known variance assumptions make those techniques even easier. But I am willing to raise my score if there are some technical novelties that I missed.

---

> ### Author Response · Authors · 2022-11-17
> **Response to Reviewer npEC**
>
> Thank you for your suggestion.
>
> **Q1**: The known variance assumptions largely reduce the difficulty of the whole problem. With this info and given the previous VOFUL algorithm which also considers the multi-level variance, this idea seems not very surprising to me.
>
> **A1**: We agree that both ML2 and VOFUL divide the whole interaction process into multiple levels, where each level consists of steps with similar reward variances. However, we would like to highlight several differences and advantages between our ML2 algorithm and the previous VOFUL algorithm. First, ML2 aims for the more general function class which only has the bounded eluder dimension assumption, while VOFUL aims for the linear function class. Second, even focusing on the linear bandit case, ML2 is computationally efficient since its constructed confidence set $\mathcal{C}_{t,l}$ is computationally tractable (it actually reduces to an ellipsoid for the linear case!). In contrast, VOFUL needs to build a test-based confidence set, which is not known to have a closed-form solution even with a known variance assumption. Thus, we believe our approach is novel even given VOFUL.
>
> ----
>
> **Q2**: Its generalized linear function results also seem a bit incremental.
>
> **A2**: To obtain a tighter confidence set in generalized linear bandits, we give a refined analysis of the regret of FTRL in Theorem 6.5, which is highly non-trivial since the previous analysis of FTRL relies on the property that all the noise can be bounded uniformly.
>
> The difference in proof techniques between ours and that of Ouhamma et al. (2021) can be highlighted as follows. For the equation (G.6), the second term was usually bounded by $\sum R^2 \\| a_s\\|\_{\Lambda_t^{-1}}^2$, and further bounded by $O(R^2 d\log t)$. In our proof, we bound this term by $\sigma_{max}^2 d$ with a high probability bound, which is nontrivial. Furthermore, the first term in equation (G.6) was usually bounded by $\sum C^2 \\| a \\|\_{\Lambda_t^{-1}}^2 \le O(C^2 d)$, which is loose. Instead, our approach bounds the $\\| a \\|\_{\Lambda_t^{-1}}^2$ with its trivial upper bound $O( \\| a \\|\_2^2)$ and treats the rest of the term as the online regret $reg_t$ itself, and it can be substituted into the original inequality to give a final tighter bound of $reg_t$. Without such proof techniques, the original regret can not be improved by a factor $d$ and a factor $R$ to $\sigma_{max}$.

---

### Official Review · Reviewer_Hrpw · 2022-10-28

**Confidence:** 4
**Clarity, Quality, Novelty And Reproducibility:** Good.
**Correctness:** 3
**Technical Novelty And Significance:** 2
**Empirical Novelty And Significance:** Not applicable
**Recommendation:** 5

**Strength And Weaknesses:**

Extending linear model to nonlinear model with provable guarantee is an important topic in bandits problems. However, the approach of using eluder dimension is a bit incremental. There is no much technical novelty as far as I know.

1. It is well known that only linear and generalized linear models have bounded eluder dimension, and the eluder dimension of one-layer neural network has exponential dependency on d. Using the Eluder dimension does not provide much more insights than linear models. So it is unclear to me how "general" the function class is.

2. The efficiency column of Table 1 is very misleading. It is also well known that UCB based on eluder dimension is computationally-inefficient for non-linear function class. I do not think an empirical risk minimization oracle is enough. If we look at line 5 in Algorithm 1, you need an optimization oracle to find f. This is a much stronger oracle.

3. Most techniques and algorithmic designs seem to be used before (variance-aware algorithm, multi-level partition scheme of the observed data). This restricted the technical novelty of the paper.

Linear Contextual Bandits with Adversarial Corruptions, NeurIPS 2021.
Nearly minimax optimal reinforcement learning for linear mixture markov decision processes. COLT 2021.

Extending the UCB-style algorithm from linear case to non-linear case using Eluder dimension also has been used in multiple papers before.

Reinforcement Learning with General Value Function Approximation: Provably Efficient Approach via Bounded Eluder Dimension. NeurIPS 2020.


**Summary Of The Paper:**

This paper studied bandits with general function class and heteroscedastic noise. A variance-dependent regret bound is derived.

**Summary Of The Review:**

This paper is a bit incremental since it combined eluder dimension approach with variance aware confidence set. Several limitations:

1. Eluder dimension provided little new information beyond linear.
2. The algorithm is not computationally efficient with non-linear function approximation.
3. The author should carefully discuss technical novelty beyond existing works since this is a pure theory paper.

---

> ### Author Response · Authors · 2022-11-17
> **Response to Reviewer Hrpw (Part 2)**
>
> **Q3**: Most techniques and algorithmic designs seem to be used before (variance-aware algorithm, multi-level partition scheme of the observed data). This restricted the technical novelty of the paper.
>
> **A3**: We believe you have overlooked the novelty of our algorithm. While variance-aware algorithms and the multi-level partition scheme have been studied/used in prior works, the specific techniques used in prior works cannot be directly applied to our setting. Therefore, we need to invent new algorithm designs and techniques to achieve variance-aware regret bounds. We highlight the key technical challenges and our new techniques as follows:
>
> 1. For the heteroscedastic noise setting, the variance-dependent regret is first achieved by Zhou et al. (2021) with the help of *Bernstein inequality for vector-valued martingale* introduced in their work. However, their Bernstein inequality only holds for the case where the variances are homoscedastic. Previous algorithms use weighted linear regression to *normalize* the variance of noises encountered at each round, which only applies to linear bandits and cannot be applied to generalized linear bandits or other general nonlinear reward functions. To overcome this problem, we propose the ML2 framework that gets around the weighted linear regression and instead partitions the observed data according to their variances followed by empirical risk minimization at each level.
>
> 2.  For the construction of the confidence sets, previous work (Zhou et al., 2021) makes use of the vector-valued Bernstein Inequality to construct an ellipsoid-style confidence set, which only works for the linear setting. We build the confidence set for generalized linear bandits by the variance-dependent online regression regret bound achieved by FTRL. This is quite different from the confidence set construction by Zhou et al., (2021).

---

> > ### Comment · Reviewer_Hrpw · 2022-12-12
> > **thanks for the response**
> >
> > Thanks for the response. I will stand on my current evaluation since the technical novelty for me is a bit limited. And if the computational efficiency can only be achieved for generalized linear models, this may lose the point of general function classes.

---

> ### Author Response · Authors · 2022-11-17
> **Response to Reviewer Hrpw (Part 1)**
>
> Thanks for your suggestions. We address your concerns separately.
>
> **Q1**: It is well known that only linear and generalized linear models have bounded eluder dimension, and the eluder dimension of one-layer neural network has an exponential dependency on d. Using the Eluder dimension does not provide much more insight than linear models. So it is unclear to me how "general" the function class is.
>
> **A1**: First, we would like to point out that the function class with bounded eluder dimension is actually strictly larger than linear and generalized linear functions, due to Thm. 6 in [1]. Next, although we consider function class with bounded eluder dimension in this paper, our main goal is not to provide a new class of general functions or a new structural complexity; instead, we aim to study how to incorporate variance information into the bandit problem when general function approximation is employed, and our proposed solutions successfully accomplish this goal. Deriving variance-dependent regret bounds for more general function classes is orthogonal to the contribution of this paper and is left as a future work.
>
> [1] Li, Gene, et al. "Eluder dimension and generalized rank." arXiv preprint arXiv:2104.06970 (2021).
>
> ----
>
> **Q2**: The efficiency column of Table 1 is very misleading. It is also well known that UCB based on eluder dimension is computationally inefficient for non-linear function class. I do not think an empirical risk minimization oracle is enough. If we look at line 5 in Algorithm 1, you need an optimization oracle to find f. This is a much stronger oracle.
>
> **A2**: We agree with your comment and have explicitly used computationally efficient and oracle-efficient to distinguish different kinds of efficiency, and also defined the oracle to include both ERM and the maximization in Line 5 of Algorithm 1. Nevertheless, we would like to emphasize that line 5 in Algorithm 1 is computationally efficient under many practical settings such as generalized linear bandit setting (See Sec. 6), where the confidence set $\mathcal{C}_{t,l}$ has a closed-form.

---

### Official Review · Reviewer_gqkL · 2022-10-30

**Confidence:** 2
**Correctness:** 3
**Technical Novelty And Significance:** 3
**Empirical Novelty And Significance:** 2
**Recommendation:** 6

**Clarity, Quality, Novelty And Reproducibility:**

The paper talks about extended models related to generalized reward function in stochastic bandit problems with good quality and clarity.


**Strength And Weaknesses:**

Strength: This paper is well written. The proposed model extends the previous generalized linear ones with sound theoretical results.

Weakness: No experimental results are provided to support the theoretical results.


**Summary Of The Paper:**

This paper considers a stochastic bandit model where the reward function belongs to a class of uniformly bounded functions and the additive noise can be heteroscedastic. A multi-level learning framework is proposed to tackle this general bandit model where this paper designs an algorithm to construct the variance-aware confidence set. For generalized linear bandits, FTRL is introduced to achieve a tighter regret.


**Summary Of The Review:**

Overall, I think that the studied model in this paper is well-motivated. The introduced variance-dependent regret and FTRL in the extended model are interesting and with sound analysis. Therefore, I recommend “weak accept”.

---

> ### Author Response · Authors · 2022-11-17
> **Response to Reviewer gqkL**
>
> Thank you for your positive comments and suggestions.
>
> While the main contribution of our paper is more on the theoretical side, we have added an experiment to validate the main theoretical results in Appendix A. In detail, we have conducted experiments on the proposed ML$^2$+FTRL algorithm for generalized linear bandits. We compare our algorithm with GLOC proposed by Jun et al. (2017). For each trial, the dimension of $\theta^*$ is set to $d = 20$, and $\theta^*$ is sampled uniformly from $[-\sqrt{1 / d}, \sqrt{1 / d}]^d$. At each round, the action set is of cardinality $100$, where each action is uniformly sampled from $[-1, 1]^d$ and normalized to a unit 2-norm vector.  The link function is set to be the sigmoid function $h(z) = \frac{1}{1 + e^{-z}}$. The variance upper bound at each round is uniformly sampled from $[0, 0.75]$. To simulate the variance-bounded condition considered in this paper, our reward is not sampled from the Bernoulli distribution. Instead, at each round $t$, we sample a random variable $\epsilon_t'$ from Poisson distribution, i.e., $\tilde{\epsilon}_t \sim Pois(\sigma_t^2)$ and $\epsilon_t' = \min\{\tilde{\epsilon}_t, R\}$ where $R$ is set to 10. Our zero-mean noise $\epsilon_t$ is computed by $\epsilon_t = \epsilon_t' - \mathbb{E}[\epsilon_t']$. With this construction, it is not hard to see that $\epsilon_t$ satisfies Assumption 3.5 with sub-Gaussian parameter $R$. From Figure 1, we can see that by adapting the multi-level and variance-aware structure, our ML$^2$ algorithm outperforms the GLOC algorithm by a large margin.
>
> We hope this experiment can help remove your concern and receive more support from you

---

> > ### Comment · Reviewer_gqkL · 2022-12-12
> > **thanks for the response**
> >
> > Thanks for the response. I will keep my initial rating after carefully reading the response.

---

### Author Response · Authors · 2022-11-18
**Follow up with Reviewer gqkL, Hrpw, npEC, pkxU**

Dear Reviewers,

Thanks for all of your helpful comments!

Since the deadline for the author-reviewer discussion phase is fast approaching, we would like to follow up with you to see if you have any further questions. In our response, we have addressed all your questions. In particular, we add experiments to corroborate our regret bound and compare it to the corresponding algorithm GLOC without our framework. In response to the reviewers' concern about the novelty of our framework, we also explain the main difficulties in our setting and the new technical tools developed in this paper to deal with them. We are looking forward to your feedback.

Best,
Authors of Paper 3544

---

### Decision · Program_Chairs · 2023-01-20

**Decision:**

Reject

**Justification For Why Not Higher Score:**

The reviewers agreed that the paper has limited novelty. Other issues raised included problems with computational complexity as well as the lack of experiments. These problems should be addressed in future versions of the papers (e.g., explaining properly and more convincingly the novel aspects would be important). While the authors provided a toy experiment in the rebuttal, at the very least it should be for the case when the variances are estimated and not given, to reduce the built-in advantage of the proposed method over variance-agnostic methods. Furthermore, a comparison with variance-aware algorithms for the linear bandit case should be included.

**Justification For Why Not Lower Score:**

N/A

**Metareview: Summary, Strengths And Weaknesses:**

The paper presents results in forms of algorithms and regret bounds for bandit learning with general function classes and heteroscedastic noise (with noise variance). While the reviewers agreed that the results are novel and sound, they raised several concerns regarding its novelty and how interesting it is for the community. As a result, unfortunately I am not able to recommend the paper for publication at this point.

**Summary Of Ac-Reviewer Meeting:**

While the paper may seem borderline from just looking at the scores, the only slightly positive reviewer had very low confidence, while all other reviewers as well as the AC shared the same concerns about novelty. As a result, no AC-reviewer meeting was held.